# Etiological, epidemiological, and clinical features of acute diarrhea in China

Li-Ping Wang[1,64], Shi-Xia Zhou [2,3,64], Xin Wang[4,64], Qing-Bin Lu [5], Lu-Sha Shi[1], Xiang Ren[1], Hai-Yang Zhang[3], Yi-Fei Wang[1], Sheng-Hong Lin[1], Cui-Hong Zhang[1], Meng-Jie Geng[1], Xiao-Ai Zhang[3], Jun Li[6], Shi-Wen Zhao[7], Zhi-Gang Yi [8], Xiao Chen[9], Zuo-Sen Yang[10], Lei Meng[11], Xin-Hua Wang[11], Ying-Le Liu[12], Ai-Li Cui[13], Sheng-Jie Lai[14,15], Meng-Yang Liu[3], Yu-Liang Zhu[1], Wen-Bo Xu[13], Yu Chen[9], Jian-Guo Wu [12], Zheng-Hong Yuan[16], Meng-Feng Li [6], Liu-Yu Huang[17], Zhong-Jie Li[1✉], Wei Liu [3,5✉], Li-Qun Fang [2,3✉], Huai-Qi Jing[4,65], Simon I. Hay [18,19,65], George F. Gao [20,65], Wei-Zhong Yang[20,65] & The Chinese Centers for Disease Control and Prevention (CDC) Etiology of Diarrhea Surveillance Study Team*

National-based prospective surveillance of all-age patients with acute diarrhea was conducted in China between 2009–2018. Here we report the etiological, epidemiological, and clinical features of the 152,792 eligible patients enrolled in this analysis. Rotavirus A and norovirus are the two leading viral pathogens detected in the patients, followed by adenovirus and astrovirus. Diarrheagenic *Escherichia coli* and nontyphoidal *Salmonella* are the two leading bacterial pathogens, followed by *Shigella* and *Vibrio parahaemolyticus*. Patients aged <5 years had higher overall positive rate of viral pathogens, while bacterial pathogens were more common in patients aged 18–45 years. A joinpoint analysis revealed the age-specific positivity rate and how this varied for individual pathogens. Our findings fill crucial gaps of how the distributions of enteropathogens change across China in patients with diarrhea. This allows enhanced identification of the predominant diarrheal pathogen candidates for diagnosis in clinical practice and more targeted application of prevention and control measures.

[1] Division of Infectious Disease, Key Laboratory of Surveillance and Early-warning on Infectious Disease, Chinese Center for Disease Control and Prevention, Beijing, China. [2] Anhui Medical University, Hefei, China. [3] State Key Laboratory of Pathogen and Biosecurity, Beijing Institute of Microbiology and Epidemiology, Beijing, China. [4] National Institute for Communicable Disease Control and Prevention, Chinese Center for Disease Control and Prevention, Beijing, China. [5] Department of Laboratorial Science and Technology, School of Public Health, Peking University, Beijing, China. [6] Sun Yat-sen University, Guangzhou, China. [7] Yunnan Center for Disease Control and Prevention, Kunming, China. [8] Shanghai Public Health Clinical Center, Shanghai, China. [9] Zhejiang University, Hangzhou, China. [10] Liaoning Provincial Center for Disease Control and Prevention, Shenyang, China. [11] Gansu Provincial Center for Disease Control and Prevention, Lanzhou, China. [12] Wuhan University, Wuhan, China. [13] National Institute for Viral Disease Control and Prevention, Chinese Center for Disease Control and Prevention, Beijing, China. [14] WorldPop, School of Geography and Environmental Science, University of Southampton, Southampton, UK. [15] Key Laboratory of Public Health Safety, Ministry of Education, School of Public Health, Fudan University, Shanghai, China. [16] Fudan University, Shanghai, China. [17] The Institute for Disease Prevention and Control of PLA, Beijing, China. [18] Department of Health Metrics Sciences, School of Medicine, University of Washington, Seattle, WA, USA. [19] Institute for Health Metrics and Evaluation, University of Washington, Seattle, WA, USA. [20] Chinese Centre for Disease Control and Prevention, Beijing, China. [64]These authors contributed equally: Li-Ping Wang, Shi-Xia Zhou, Xin Wang. [65]These authors jointly supervised this work: Huai-Qi Jing, Simon I. Hay, George F. Gao, Wei-Zhong Yang. *A list of authors and their affiliations appears at the end of the paper. ✉email: lizj@chinacdc.cn; liuwei@bmi.ac.cn; fang_lq@163.com

D iarrhea remains one of the major causes of disease burden worldwide, despite significant progress in sanitation status and public health awareness[1–3]. More than 4.4 billion cases and 1.6 million deaths (ranking eighth as a common cause of life loss) due to diarrhea occur worldwide in 2016[1], causing substantial medical and healthcare costs and a high economic impact on society. Evidence shows that diarrheal disease is a major contributor to pediatric morbidity and mortality, especially in low-income and middle-income countries (LMICs)[4]. More than 0.5 million of diarrheal deaths occurred among children younger than 5 years globally in 2017, 88% of which occurred in South Asia and sub-Saharan Africa[5]. The etiology of acute diarrhea differs between regions depending on economic development, local climate, and geography. A better understanding of the epidemiology, etiology, and seasonality of acute diarrhea would be valuable for planning and adopting targeted preventive measures, as well as antimicrobial therapy.

China is one of the 15 countries with a high incidence of diarrhea[6]. Although there are numerous studies that investigated the gastrointestinal pathogens in acute diarrhea[7–9], the previous studies were subject to limitations such as small sample size, limited coverage area, short surveillance duration, few tested pathogens, and small catchment populations. In addition, pooled data analysis was hindered by inconsistent detection assays among studies. Here based on a national surveillance network for patients with acute diarrhea, we made the systematic attempt to identify the etiological, epidemiological, and clinical features of acute diarrhea in an all-age population for a decade in China. It is anticipated that long-term continuous collection of surveillance data will be representative of a wide range of patients valuable for planning and adopting targeted preventive measures and therapy.

## Results

From January 2009 to December 2018, 157,883 patients with acute diarrhea were recruited, from whom 5091 patients were excluded due to incomplete data or not initially diagnosed in the sentinel hospitals, thus 152,792 patients were used for the final analysis (Fig. 1). There were 85,731 patients tested for at least one virus, 100,129 patients tested for at least one bacterium, and 12,988 patients tested for at least one parasite. From those, 58,620 patients were tested for all seven viruses, 59,384 patients were tested for all 13 bacteria, 11,167 patients were tested for all three parasites, and 3330 patients were tested for all 23 pathogens. The demographic and epidemiological characteristics of the studied patients are shown in Table 1. Of these, 58.96% (90,093/152,792) were male, 55.63% (85,001/152,792) were aged <18 years and 11.02% (16,847/152,792) were older people aged ≥60 years. A total of 116,217 patients (76.06%) were from urban areas. The definite outcome was reported from 89,441 patients, of whom 40 patients (0.04%) died. On average, 15,536 (interquartile range: 9592–18,065) patients from 126 (interquartile range: 91–149) hospitals were analyzed annually (Supplementary Fig. 1). There were 26.07% (868/3330) of patients who were positive for at least one enteropathogen (Supplementary Table 1). These patients were younger and took longer from onset to hospital admission compared with negative patients (median age, 1 vs 35 years, delay 3 vs 1 day) (Supplementary Table 2).

**The pathogen spectrum based on 58,620 patients with all seven viruses tested**. In total, 58,620 cases were tested for all the seven viruses, from whom comparable demographic characteristics were recorded (Table 1). At least one positive detection was obtained from 18,676 (31.86%) cases. Rotavirus A showed the highest proportion of positive detection (45.05%), followed by

norovirus (32.21%) and adenovirus (9.27%) (Fig. 2a and Supplementary Table 3).

The pathogen spectrum differed across age groups. For pediatric patients <18 years old, the top three viral pathogens were rotavirus A, norovirus, and adenovirus. The ranking differed among pediatric cohorts. The leading pathogen in the age group <5 years was rotavirus A, replaced by norovirus in the 5–17 age group. In adults (≥18 years old), norovirus remained the leading pathogen, followed by rotavirus A and astrovirus (Fig. 2a and Supplementary Table 3).

**The pathogen spectrum based on 59,384 patients with all 13 bacteria tested**. In total, 59,384 cases were tested for all 13 bacterial pathogens, from which comparable demographic characteristics were collected (Table 1). At least one positive detection was obtained from 10,825 (18.23%) cases. The leading bacterial pathogen was diarrheagenic *Escherichia coli* (DEC) (43.31% of all positive detection), followed by nontyphoidal *Salmonella* (NTS) (25.11%), *Vibrio parahaemolyticus* (*V. parahaemolyticus*) (10.83%), and *Shigella* (6.88%) (Fig. 2b and Supplementary Table 3).

The pathogen spectrum differed among age groups. Overall, DEC and NTS were the leading bacterial pathogens that were identified from pediatric patients <18 years old. For those aged <6 months, the top five bacteria were DEC, NTS, *Aeromonas hydrophila* (*A. hydrophila*), *Shigella*, and *V. parahaemolyticus*, which was altered to NTS, DEC, *A. hydrophila*, *Shigella* and *Campylobacter jejuni* (*C. jejuni*) in the 6–11 months group. For the 1–4 years group, the top five bacteria were DEC, NTS, *Shigella*, *C. jejuni* and *A. hydrophila*, with the same top four pathogen in the 5–17 age group, while 5th was replaced by *V. parahaemolyticus*. In the three adult groups (18–45, 46–59, and ≥60 years old), the top three pathogens were DEC, NTS, *V. parahaemolyticus* (Fig. 2b and Supplementary Table 3).

**The pathogen spectrum based on 11,167 patients with all three parasites tested**. In total, 11,167 cases were tested for all three parasites pathogens. The leading parasitical pathogen was *Entamoeba histolytica* (*E. histolytica*) (55.38% of all positive detection), followed by *Giardia lamblia* (*G. lamblia*) (32.82%) and *Cryptosporidium* (11.79%). For the parasitical pathogen spectrum, high consistency in the ranking was observed among the various age groups of adults (Fig. 2c and Supplementary Table 3).

**Detection of specific viral pathogens**. The frequency of testing for each of the seven viral pathogens differed from 71,567 for rotavirus C to 80,054 for adenovirus (Supplementary Table 4). The highest positive rate was to rotavirus A (20.40%, 15,155/74,307), followed by norovirus (12.47%, 9,707/77,855), adenovirus (3.33%, 2669/80,054), and astrovirus (2.77%, 2201/79,529) (Fig. 3a and Supplementary Table 4). The same ranking was obtained for all viruses-tested specimens (Supplementary Table 3).

Across the study period, rotavirus A remained as the most frequently identified virus (Supplementary Fig. 2), however, the positive rate declined from 39.59% in the first year to 14.61% in the last year, showing a significantly decreasing trend with an average annual percentage change (AAPC) of −12.4% ($p < 0.01$) (Supplementary Table 5). A biennial pattern emerged, with alternating years of low and high rotavirus activity. Sapovirus, conversely, showed an increasing trend with an AAPC of 6.6% ($p < 0.01$). For rotavirus A positive specimens, 50.17% (7,604/15,155) were successfully genotyped, with the most prevalent genotype as G9P[8] (14.69%), followed by G3P[8] (8.56%) and G1P[8] (6.83%) (Supplementary Table 6). GII was the most

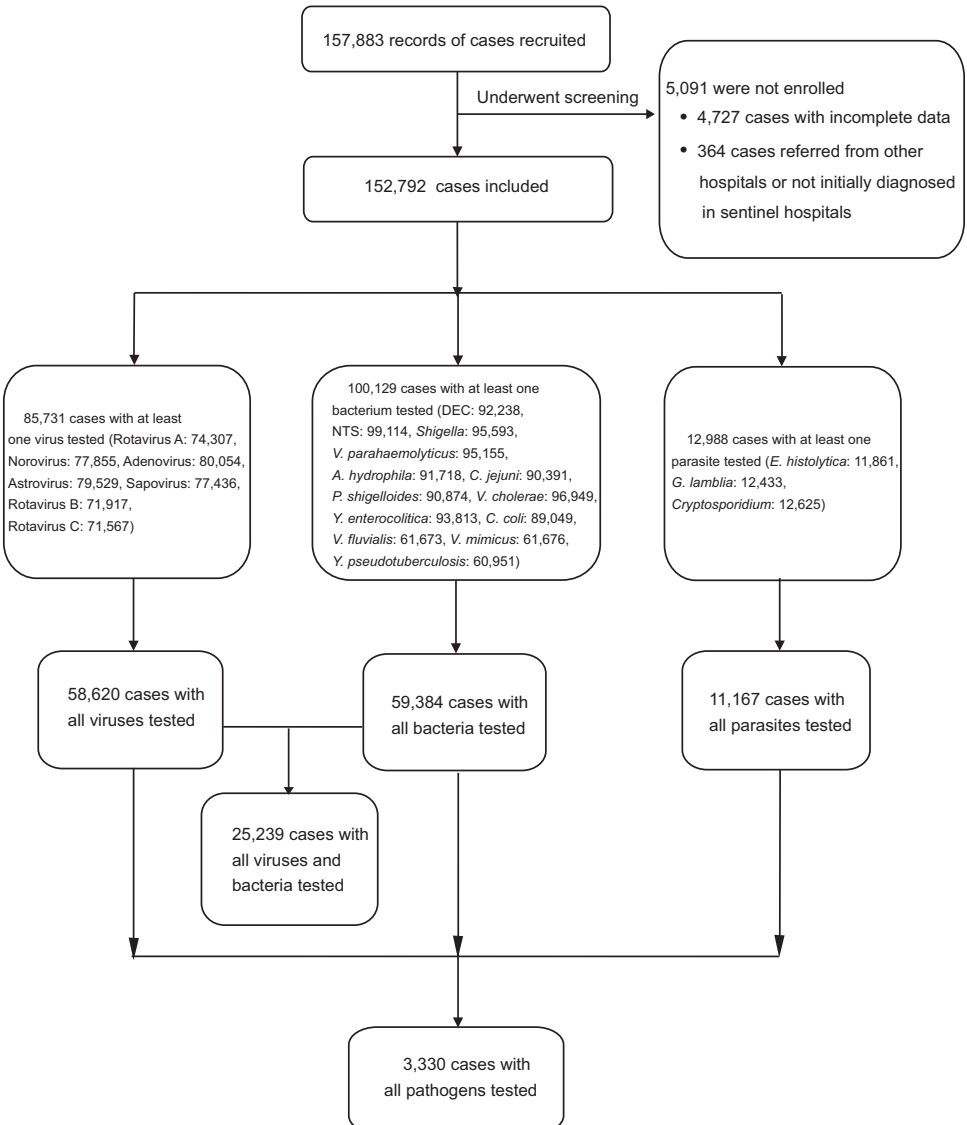

**Fig. 1 A flowchart of the data collection and sorting procedures.** This flow diagram summarizes the number of patients with acute diarrhea for each analysis in this study.

common genotype of norovirus 90.85% (8426/9275) (Supplementary Table 4).

Based on the seven age group classifications, 46.73% (6874/14,710) of the children aged 6–11 months had at least one positive viral detection, dropping to 27.85% (1361/4887) in 5–17 year adolescents, 21.55% (3734/17,324) in 18–45 year, 21.86% (1607/7352) in 46–59 year and 19.82% (1711/8633) in ≥60 years. All viruses were detected with the highest frequency in children (<5 years old), except for norovirus, with the highest frequency seen in 18–45 year adults ($p < 0.05$) (Fig. 3a and Supplementary Table 4). Joinpoint regression analysis revealed a similar trend for rotavirus A and B, adenovirus, norovirus; all showing one descending turnaround point at 2–3 years old, although norovirus had an additional turnaround point at 5 years, revealing an obvious increase. The stabilized or decreased turnpoints were shown at around 21 years old for all these viruses (Fig. 4a). Rotavirus, adenovirus, and sapovirus were detected with slightly higher rates in males (all $p < 0.05$) (Supplementary Table 4). Among patients under 5 years of age, only the rate for rotavirus was significant for sex ($p < 0.001$) (Supplementary Table 7). Astrovirus was more frequently

identified among patients living in rural areas than those living in urban areas ($p < 0.0001$) (Supplementary Table 8).

**Detection of specific bacterial pathogens**. The frequency of testing for each of the 13 bacterial pathogens differed from 60,951 for *Yersinia pseudotuberculosis* (*Y. pseudotuberculosis*) to 99,114 for NTS (Supplementary Table 4). Among the 13 tested bacterial pathogens, DEC had the highest positive rate (6.71%, 6191/92,238), followed by NTS (4.41%, 4366/99,114), *Shigella* (2.44%, 2331/95,593), and *V. parahaemolyticus* (2.08%, 1981/95,155) (Fig. 3b and Supplementary Table 4). Of the DEC positive specimens, enteroaggregative *E. coli* was most frequently detected (28.51%), followed by enteropathogenic *E. coli* (25.36%), enterotoxigenic *E. coli* (24.07%), enteroinvasive *E. coli* (6.59%), and enterohaemorrhagic *E. coli* (3.02%).

Across the study years, four bacterial pathogens showed significantly increasing trends of positivity, with extraordinarily high AAPC for *Campylobacter coli* (*C. coli*) and *C. jejuni* (17.5% and 14.5%, respectively) (Supplementary Fig. 2 and Supplementary Table 5). In contrast, *Shigella*, *Vibrio cholerae* (*V. cholerae*)

**Table 1 Demographic and epidemiological characteristics of patients with diarrhea in the mainland of China, 2009–2018.**

| | All cases | Any virus tested | Any bacterium tested | Any parasite tested | All viruses tested | All bacteria tested | All parasites tested | All pathogens tested |
|---|---|---|---|---|---|---|---|---|
| Total (N) | 152,792 | 85,731 | 100,129 | 12,988 | 58,620 | 59,384 | 11,167 | 3330 |
| **Sex N (%)** | | | | | | | | |
| Male | 90,093 (58.96) | 50,695 (59.13) | 57,409 (57.34) | 7510 (57.82) | 34,223 (58.38) | 33,248 (55.99) | 6538 (58.55) | 1970 (59.16) |
| Female | 62,699 (41.04) | 35,036 (40.87) | 42,720 (42.66) | 5478 (42.18) | 24,397 (41.62) | 26,136 (44.01) | 4629 (41.45) | 1360 (40.84) |
| **Age N (%)** | | | | | | | | |
| <5 year | 75,687 (49.54) | 47,535 (55.45) | 39,697 (39.65) | 4416 (34.00) | 30,393 (51.85) | 24,061 (40.52) | 4027 (36.06) | 1059 (31.80) |
| <6 month | 17,673 (11.57) | 10,820 (12.62) | 9544 (9.53) | 873 (6.72) | 6779 (11.56) | 5711 (9.62) | 828 (7.41) | 186 (5.59) |
| 6–11 month | 23,201 (15.18) | 14,710 (17.16) | 11,682 (11.67) | 1251 (9.63) | 9632 (16.43) | 7149 (12.04) | 1122 (10.05) | 311 (9.34) |
| 1–4 year | 34,813 (22.78) | 22,005 (25.67) | 18,471 (18.45) | 2292 (17.65) | 13,982 (23.85) | 11,201 (18.86) | 2077 (18.60) | 562 (16.88) |
| 5–17 year | 9314 (6.10) | 4887 (5.70) | 6018 (6.01) | 774 (5.96) | 3169 (5.41) | 3570 (6.01) | 690 (6.18) | 222 (6.67) |
| 18–45 year | 35,640 (23.33) | 17,324 (20.21) | 28,627 (28.59) | 3751 (28.88) | 12,997 (22.17) | 16,135 (27.17) | 3130 (28.03) | 996 (29.91) |
| 46–59 year | 15,304 (10.02) | 7352 (8.58) | 12,341 (12.33) | 1782 (13.72) | 5621 (9.59) | 7203 (12.13) | 1444 (12.93) | 396 (11.89) |
| ≥60 year | 16,847 (11.02) | 8633 (10.07) | 13,446 (13.43) | 2265 (17.44) | 6440 (10.99) | 8415 (14.17) | 1876 (16.80) | 657 (19.73) |
| **Ecological regions N (%)** | | | | | | | | |
| Northeast China | 7971 (5.22) | 3741 (4.36) | 5664 (5.66) | 347 (2.67) | 3633 (6.20) | 2687 (4.52) | 346 (3.10) | – |
| North China | 35,962 (23.54) | 12,836 (14.97) | 25,436 (25.40) | 1709 (13.16) | 9499 (16.20) | 17,667 (29.75) | 1362 (12.20) | 230 (6.91) |
| Inner Mongolia-Xinjiang | 9577 (6.27) | 3732 (4.35) | 7635 (7.63) | 2 (0.02) | 2285 (3.90) | 2441 (4.11) | 2 (0.02) | – |
| Qinghai-Tibet | 1006 (0.66) | 853 (0.99) | 967 (0.99) | – | 740 (1.26) | 316 (0.53) | – | – |
| Southwest China | 6939 (4.54) | 4236 (4.94) | 4950 (4.94) | 1773 (13.65) | 4117 (7.02) | 2602 (4.38) | 1534 (13.74) | 1194 (35.86) |
| Central China | 73,176 (47.89) | 49,095 (57.27) | 45,988 (45.93) | 8627 (66.42) | 31,200 (53.22) | 28,467 (47.94) | 7463 (66.83) | 1882 (56.52) |
| South China | 18,161 (11.89) | 11,238 (13.11) | 9489 (9.48) | 530 (4.08) | 7146 (12.19) | 5204 (8.76) | 462 (4.14) | 24 (0.72) |
| **Residence N (%)[a]** | | | | | | | | |
| Urban | 116,217 (76.06) | 68,774 (80.22) | 75,113 (75.02) | 11,094 (85.42) | 45,706 (77.97) | 44,495 (74.93) | 9368 (83.89) | 2243 (67.36) |
| Rural | 29,707 (19.44) | 13,542 (15.8) | 20,389 (20.36) | 1601 (12.33) | 9799 (16.72) | 11,297 (19.02) | 1519 (13.6) | 849 (25.5) |
| **Outcome N (%)** | 89,441 (58.54) | 54,198 (63.22) | 62,162 (62.08) | 7078 (54.50) | 40,050 (68.32) | 57,339 (96.56) | 6650 (59.55) | 3264 (98.02) |
| Death | 40 (0.04) | 20 (0.04) | 26 (0.04) | 6 (0.08) | 16 (0.04) | 23 (0.04) | 5 (0.08) | 1 (0.03) |

Data are presented as the N, the case number (% as proportion), unless otherwise indicated. Percentages may not total 100 because of rounding.
[a]The number of patients in rural and urban areas did not equal the total number of patients due to missing data.

and *A. hydrophila* showed significantly decreasing positivity, with AAPC of −26.5%, −26.0%, and −14.2%, respectively.

Based on seven age groups classification, the children <6 months had the lowest rate for at least one positive bacterial detection (7.92%, 756/9544), increased to the highest rate in 18-45 years (20.27%, 5802/28,627), and decreased thereafter (Supplementary Table 4). For each detected bacterium, the age-specific pattern differed. Joinpoint regression analysis showed a similar trend for NTS, *Shigella*, *C. jejuni,* and *Yersinia enterocolitica* (*Y. enterocolitica*), which were more frequently identified in children, with similar turnaround point seen at 2–5 years coincident with a decreased annual percent change (APC) (Fig. 4b). DEC, *V. parahaemolyticus*, *Plesiomonas shigelloides* (*P. shigelloides*) and *V. cholerae* were more frequently identified in young adults aged from 19-28 years, with both *P. shigelloides* and *V. cholerae* showing obvious descending turnaround points at 19-24 years old, and a second stabilized point at around 27 years old (Fig. 4b). Sex-specific differences were observed for DEC and *V. parahaemolyticus* only, with higher frequency observed in females than in males. *Shigella* was more frequently identified among patients living in rural areas than those living in urban areas (Supplementary Table 8).

**Detection of specific parasitical pathogens**. The frequency of testing for each of the three parasitical pathogens differed from 11,861 for *E. histolytica* to 12,625 for *Cryptosporidium* (Supplementary Table 4). *E. histolytica* had the highest positive rate (1.05%, 125/11,861), followed by *G. lamblia* (0.78%, 97/12,433) and *Cryptosporidium* (0.27%, 34/12,625). *E. histolytica* and *G. lamblia* showed significantly decreasing trends of positivity across the study years, with AAPC of −37.4% (−49.5 to −22.5), −30.5% (−40.7 to −18.5) (Supplementary Table 5). *Cryptosporidium* was detected with higher frequency in male ($p < 0.05$). *E. histolytica* and *G. lamblia* were detected with higher frequencies in the adult, attaining the highest rate in 46-59 years for *E. histolytica* and 18-45 years old for *G. lamblia* ($p < 0.0001$) (Supplementary Table 4).

**Co-infection pattern**. Co-infection rate among viruses was 3.03% (1779/58,620), higher than that among bacteria (0.90%, 536/59,384), and among parasites (0.03%, 3/11,167) (Fig. 2). Children had a higher viral-viral co-infection rate, and within this group, the highest level was seen in children of 1-4 years old ($\chi^2 = 686.83$, $p < 0.0001$). In contrast, adults of 18-45 years old had the highest bacterial-bacterial co-infection rate ($\chi^2 = 74.01$, $p < 0.0001$). For those with all viruses-tested patients, we observed positive correlations between norovirus–adenovirus, norovirus–astrovirus, adenovirus–astrovirus, adenovirus–sapovirus, astrovirus–sapovirus, all with an odds ratio (OR) >1 and $p < 0.05$ adjusted by Holm's method. We observed negative correlations between rotavirus–norovirus, rotavirus–adenovirus, rotavirus–sapovirus, norovirus–sapovirus at the individual level, all with OR < 1 with p-values < 0.05 adjusted by Holm's method (Supplementary Fig. 3c).

Among 25,239 patients tested for all seven viruses and 13 bacteria, 2.63% (665/25,239) had viral-bacterial co-infection (Supplementary Fig. 3a), with primarily occurred among rotavirus A, norovirus, adenovirus, sapovirus, astrovirus, DEC, and NTS (Supplementary Fig. 3b).

**Clinical syndrome and signs**. Based on 8758 patients with positive detection, different clinical syndrome and clinical signs were observed between patients with a single viral infection or single bacterial infection. More vomiting ($z = 14.05$, $p < 0.001$), respiratory symptoms ($z = 6.79$, $p < 0.001$), neurological symptoms ($z = 3.19$, $p < 0.01$), and less fever ($z = −5.58$, $p < 0.001$) were exhibited in those patients with single virus infections. These

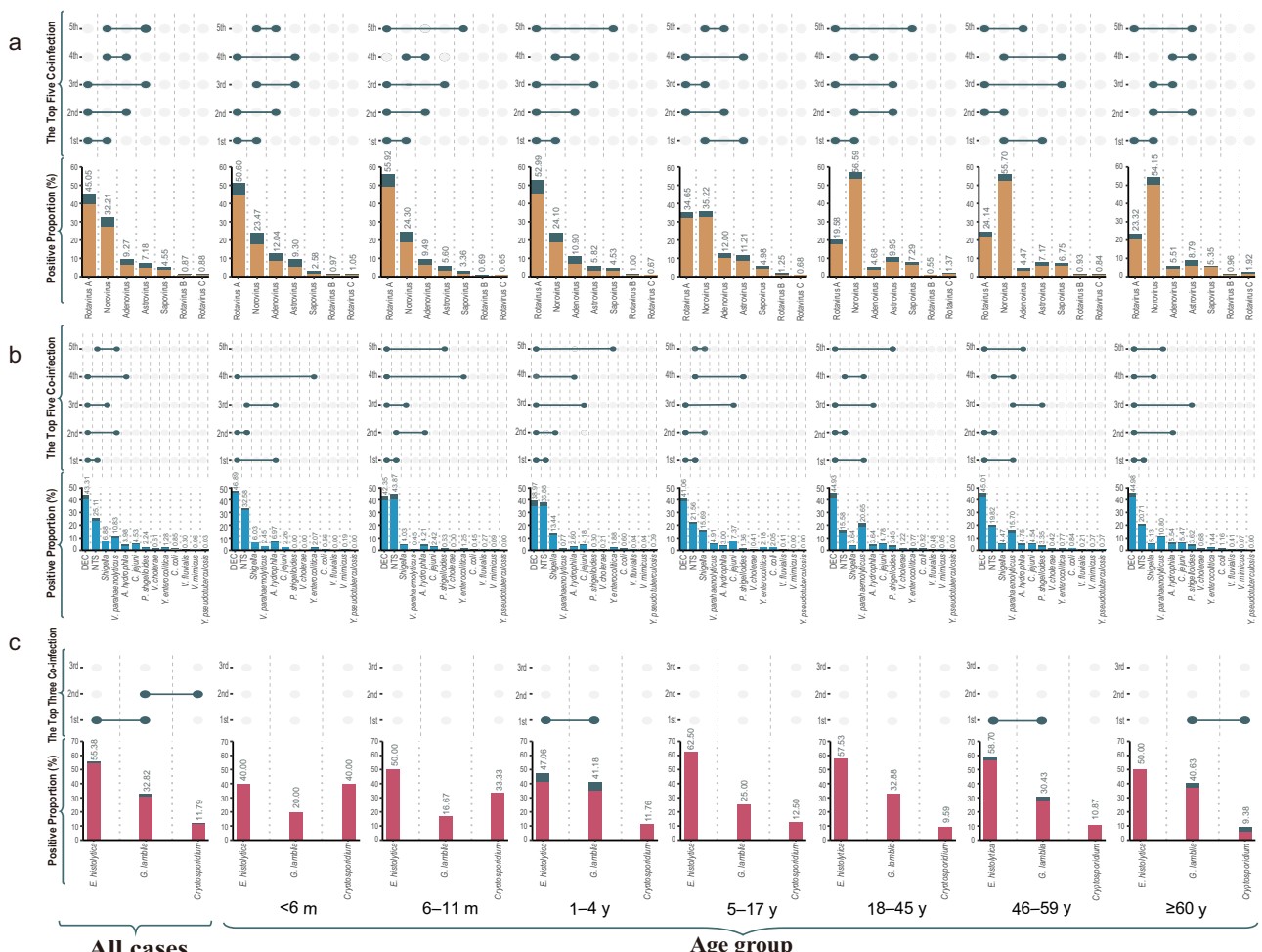

**Fig. 2 Pathogen spectrum and co-infection pattern of enteropathogens in diarrhea patients by age group in the mainland of China, 2009–2018. a** The viral spectrum and co-infection pattern of 58,620 patients with acute diarrhea who had all the seven viral pathogens tested. **b** The bacterial spectrum and co-infection pattern of 59,384 patients with acute diarrhea who had all the 13 bacterial pathogens tested. **c** The parasitical spectrum and co-infection pattern of 11,167 patients with acute diarrhea who had all three parasitical pathogens tested. The proportion of each positive pathogen was noted in % and by the length of colored bars. The orange bar indicates viral mono-infection; the blue bar indicates bacterial mono-infection; the red bar indicates parasitical mono-infection; the green bar indicates co-infection. For viruses, co-infection means co-infection among viruses. For bacteria, co-infection means co-infection among bacteria. For parasites, co-infection means co-infection among parasites. DEC diarrheagenic *Escherichia coli*; NTS nontyphoidal *Salmonella*. Source data are provided as a Source Data file.

intergroup differences were consistently observed for children and adults (Supplementary Table 9). To attain a differential diagnosis between bacterial and viral infections with acute diarrhea, a binary eXtreme Gradient Boosting model was applied on 5816 patients with viral single infections and 2942 patients with bacterial single infections. In total, 11 valid variables regarding demographical and clinical symptoms/syndromes were entered into the model, with the season, age, mucous stool, vomiting, respiratory symptoms, mushy stool, bloody stool, watery stool, sex, fever, and neurologic symptoms observed. Among them, the season was the most important predictor with a relative contribution of 37.46%, followed by age (16.73%). The contributions of mucous stool, vomiting, and respiratory symptoms were slightly larger than 5%, and the contribution for each of the others was less than 5%. The area under curve of the model was 0.79 [95% confidence interval (CI): 0.77–0.81], accuracy = 74.14%. (Supplementary Fig. 4a, b). The mean accuracy of cross-validation was 74.52% and the variance was 4.17%. Based on this multivariate logistic regression model, child patients with acute diarrhea, vomiting, and respiratory symptoms that occurred in the cold season were shown to be likely caused by virus infections,

while an adult with mucous or bloody stool, that occurred in warm-season were probably infected with bacteria (Supplementary Table 10).

**The spatial-temporal pattern**. The geographic diversity of detection and the seasonal patterns for each pathogen were demonstrated for seven ecological regions (Fig. 5). The effect on the spatial and the seasonal pattern was explored at the sentinel city level by using four sociological and six meteorological factors (Supplementary Table 11). The daily mean temperature affected most of the detected viruses and bacteria significantly, but with different directions of effects. Incidence rate ratios (IRRs) for viruses ranged from 0.83 (95% CI: 0.82–0.84) to 0.96 (95% CI: 0.94–0.97), while IRRs for bacteria ranged from 1.26 (95% CI: 1.23–1.28) to 1.70 (95% CI: 1.62–1.78). Consistent with these effects, the viral infection exhibited a winter-spring seasonality that spanned from December–January and ending in April–May, while the bacterial infection exhibited a Summer–Autumn seasonality that spanned from May to October with peaks in summer (Fig. 5). At the city level, the proportion of children significantly affected by viral infection was greater, with the IRRs ranging from 1.03 (95%

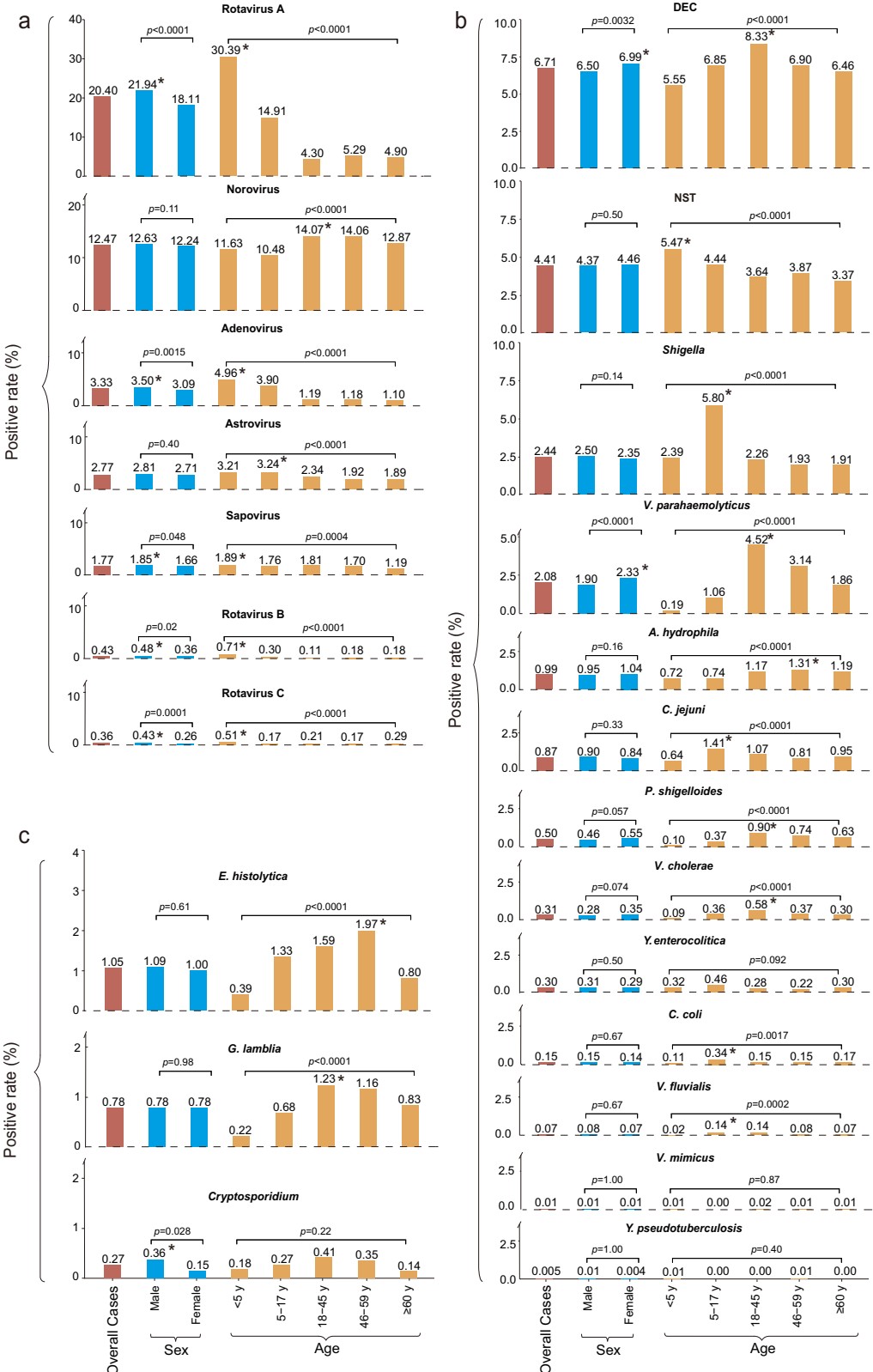

**Fig. 3 Positive rate of pathogens by sex and age in patients with diarrhea in China, 2009–2018. a** Viral pathogens. **b** Bacterial pathogens. **c** Parasite pathogens. The lengths of red bars indicate the positive rate of each pathogen. The lengths of blue bars and yellow bars indicate the positive rate of each pathogen by sex and age groups. The same group was marked by the same colors of bars filling. *Indicates a significant difference (chi-square test or Fisher's exact test were two-sided and *p* < 0.05 was statistically significant) found within the group. DEC diarrheagenic *Escherichia coli*; NTS nontyphoidal *Salmonella*. Source data are provided as a Source Data file.

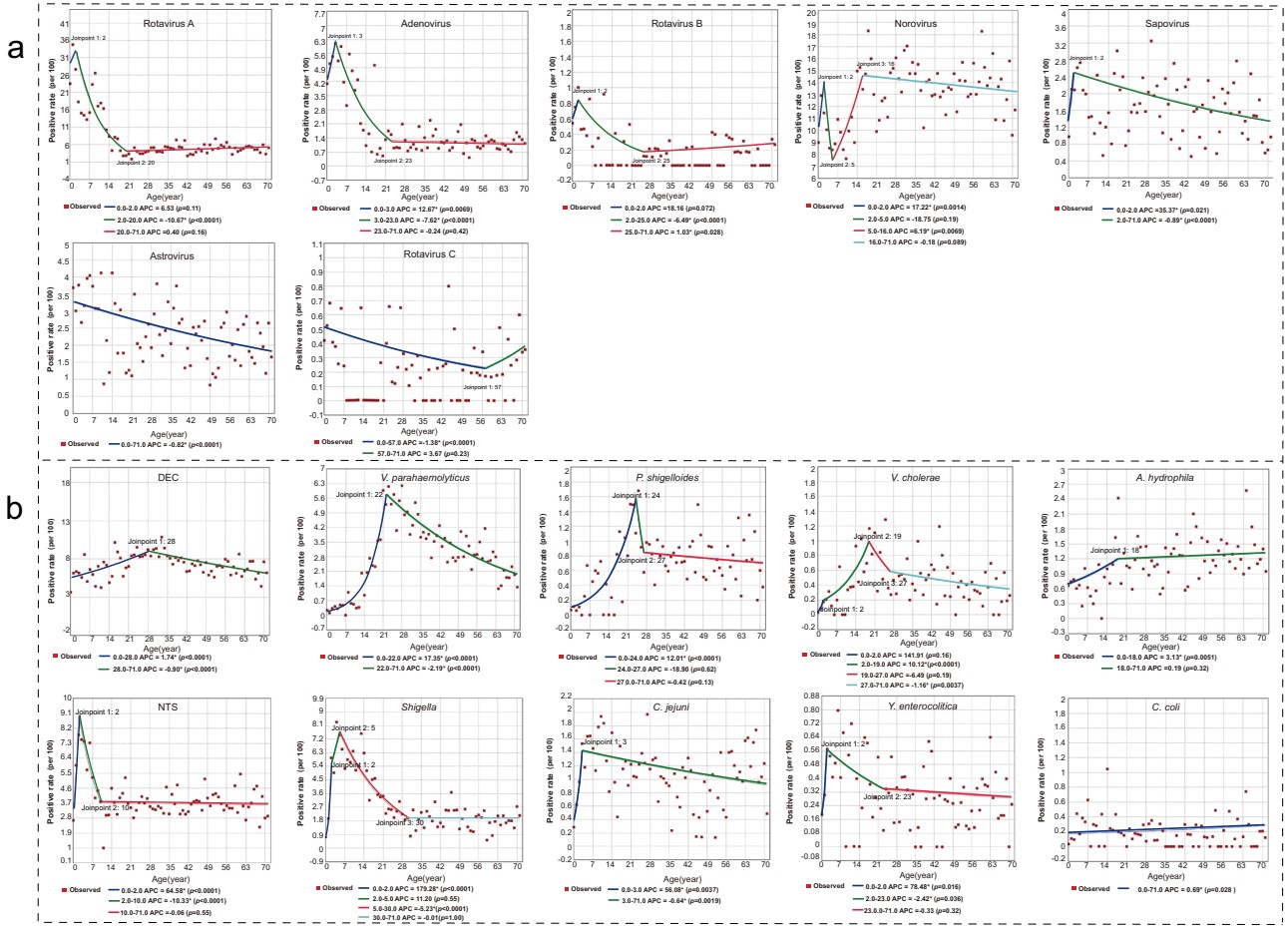

**Fig. 4 The joinpoint regression of the positive rates of each enteropathogen by age of the patient. a** Viral pathogens. **b** Bacterial pathogens. A red point indicates the positive rate of patients in terms of age and the colored curves indicate fitted patterns by the red points. Legends give the annual percentage change (APC) value of each fitted curve for each enteropathogen. *Indicates that the APC is significantly different from zero at $p < 0.05$. A normal ($Z$) distribution was used to assess the significance of the APC. All statistical tests were two-sided and $p < 0.05$ was statistically significant. DEC diarrheagenic *Escherichia coli*; NTS nontyphoidal *Salmonella*. Source data are provided as a Source Data file.

CI: 1.02–1.04) to 1.17 (95% CI: 1.13–1.21). The population density was more important for bacterial infection, the IRR of *C. jejuni* was 2.44 (95% CI: 2.26–2.65) (Supplementary Table 11).

The direction and magnitude of the evaluated effects revealed two viral clusters and two bacterial clusters. Norovirus, sapovirus, astrovirus, and adenovirus constituted Cluster I, with the positive rates of the former three positively associated with lower temperature, and positive rates of latter two negatively associated with higher wind speed (Supplementary Table 11 and Fig. 5e). Rotavirus A, B, and C constituted Cluster II, all were positively associated with a lower sunshine hour, wind speed, and a higher proportion of children. DEC, *Shigella*, NTS and *A. hydrophila* constituted Cluster III, all were associated with higher temperature, while *Shigella* and *A. hydrophila* additionally associated with a lower gross domestic product (GDP) *per capita* (Fig. 5c and Supplementary Table 11). *C. jejuni* and *C. coli* constituted Cluster IV, both positively associated with high population density (Fig. 5f).

The seasonal patterns of enteropathogens also differed across geographical locations. For example, rotavirus was strongly seasonal in the temperate areas with latitude above 40°, where epidemics peaked during the winter. In contrast, its seasonality was less pronounced in the subtropical and tropical regions, characterized by multiple peaks in the regions with latitude below 40°. The seasonality of norovirus shows an obvious bimodal pattern, and in a similar pattern as rotavirus, the peaking time in

areas with latitude below 30° or between 30° and 40° was shown earlier than that regions with latitude above 40° (Supplementary Fig. 5).

## Discussion

In this study of longitudinal surveillance spanning ten years in China, we provided updated results on the viral, bacterial, and parasitical etiologies in patients with acute diarrhea which differed in terms of patients' demography, epidemic season, and socioeconomic level.

The effect of age on pathogen detection was explored. Generally, younger patients, especially those <5 years old, had a higher frequency and variety of viral infections, while adults of 18-45 years were more likely to be infected with bacterial pathogens. Within the pediatric group, the peak detection level for each virus varied. By performing joinpoint regression analysis, we could infer the positive rate of rotavirus, norovirus, and adenovirus peaking at the age of 2–3 years old and decreasing thereafter. For norovirus, a second peak at 16 years old was observed, marking the significant turnpoint at which the trend switched. Three years old and 16 years old are the thresholds when children enter kindergarten and high school, respectively in China, possibly leading to the establishment of the host protective immunity and the ensuing decreased infection after this age. In this study, rotavirus was found to be the predominant pathogen

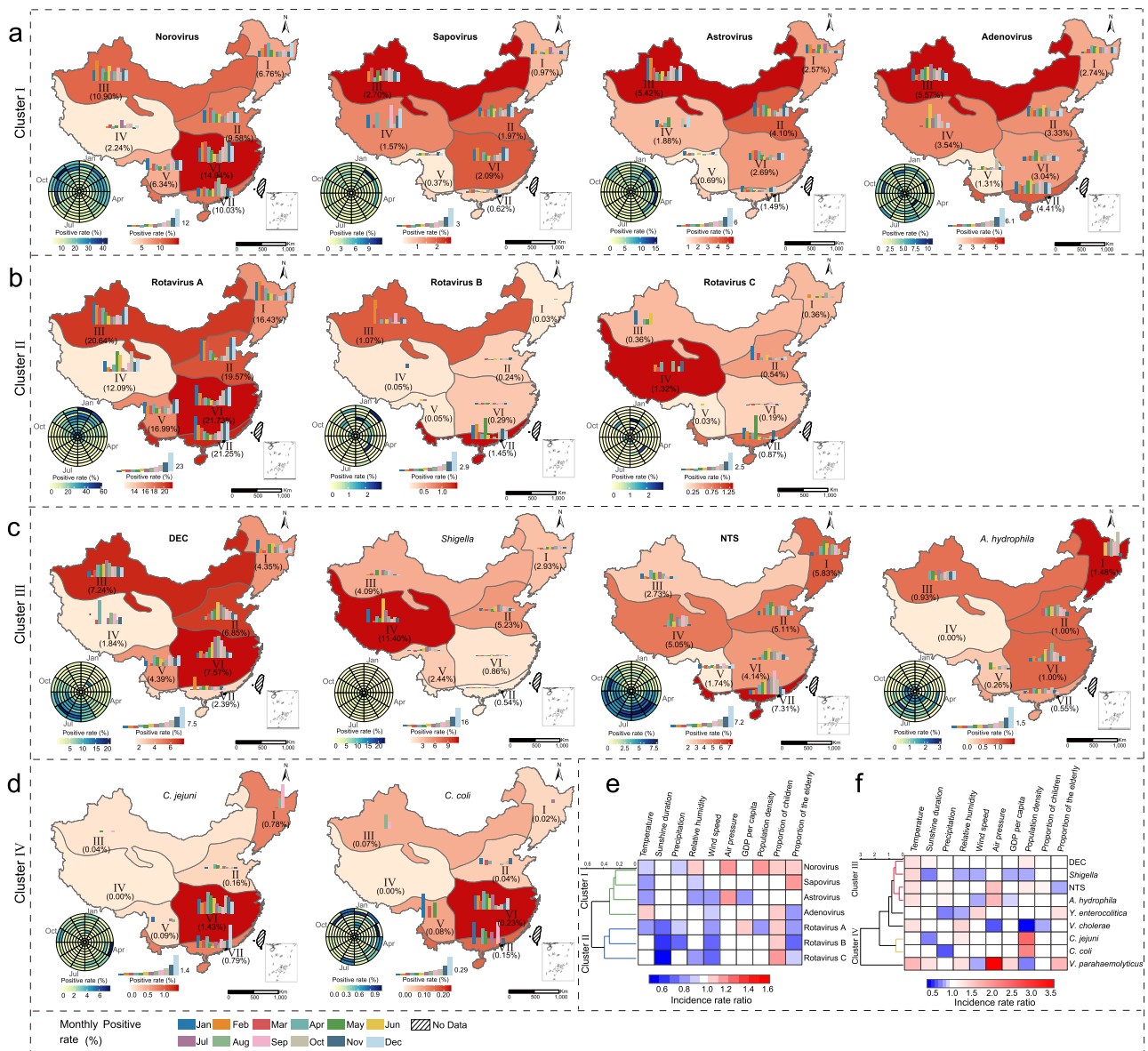

**Fig. 5 Spatial and temporal pattern of four clusters of enteropathogens in patients with diarrhea in the mainland of China during 2009–2018.** Thirteen of the tested enteropathogens were formed into four clusters (**a–d**). The seasonality was presented with a radar diagram based on the monthly positive rate from 2009 to 2018. The circumference is divided into 12 months in a clockwise direction, and the radius from inside to outside represents a particular year from 2009 to 2018. Seven ecological regions were marked (I, Northeast China district; II, North China district; III, Inner Mongolia-Xinjiang district; IV, Qinghai-Tibet district; V, Southwest China district; VI, Central China district; VII, South China district). Within each ecological region, the positive rate was noted and reflected by the background color. Histograms within each ecological region indicate the monthly positive rate averaged from 10-year data. The height of the bars indicates the positive rate of each month. *No sample in the month. The dendrogram in **e**, **f** display the clusters I–IV. Green lines mean cluster I. Blue lines mean cluster II. Red lines mean cluster III. Yellow lines mean cluster IV. The incidence rate ratio, ranging from negative influence (blue) to positive influence (red). The features used for clustering are variables with statistically significant differences in the negative binomial model. DEC diarrheagenic *Escherichia coli*; NTS nontyphoidal *Salmonella*. Source data are provided as a Source Data file.

overall, also the leading viral pathogen among children, while norovirus was the leading pathogen in adults. This is consistent with the previous findings[10]. It's noteworthy that the leading viral pathogen switched from rotavirus among children to norovirus among adults, with the high detection level maintained in all-age adults. This could be in part a function of the continuous mutation and recombination abilities of norovirus, generating novel strains with a high potential of causing outbreak events and sporadic cases[11].

In contrast with a viral infection, divergent turnpoints of age were presented for the detection of bacterial pathogens. NTS, *Shigella*, *C. jejuni* and *Y. enterocolitica* displayed a "Child-

Pattern", with a detection turnpoint seen at 2–5 years, while DEC, *V. parahaemolyticus*, *P. shigelloides* and *V. cholerae* displayed an "Adult-Pattern", with turnpoints seen at 19–28 years. Consistent with previous findings, the patients aged ≥60 years were the least likely to get infectious diarrhea[12]. This diversity of age distribution might reflect a natural change in host immunity[13] and/or dietary habits that are related to age. These findings of the effect of age on the pathogen detection might provide scientific evidence for finding the optimal timing to enhance prevention measures for each pathogen, e.g., for the diagnosis and prevention of norovirus, once a vaccine is available, it should be a priority in the 2–3 year age group, and additionally at the age of 15–16 years.

We found a slow decrease in the detection of rotavirus across the study years, probably owing to the rotavirus vaccine interventions that had been advocated after the year of 2000[14]. This was in also line with previous results showing a similar decreased rate of rotavirus during post-rotavirus vaccine introduction in Coastal Kenya[15]. Rotavirus vaccine was not mandatory in China, but higher Lanzhou lamb rotavirus vaccination coverage might indeed decrease the incidence risk among children younger than 4 years according to the city-level ecological study[16]. In many countries with universal vaccination for rotavirus, the childhood deaths of rotavirus have decreased significantly[17]. More than 28,000 deaths among children younger than five years were estimated to have been averted by the usage of the rotavirus vaccine, and its expanded use, particularly in sub-Saharan Africa, could have prevented about 20% of all deaths attributable to acute diarrhea among children younger than five years in 2016[17]. The current findings supported the potential benefit of including rotavirus vaccine into the schedule of immunization of infants in China. Consistent with a previous study[18], we found the biennial circulation patterns of rotavirus might be caused by an increase in the number of young, unexposed persons during years of low circulation, which leads to a larger number of susceptible persons acquiring and transmitting the infection in the following year. Decreased detection of several bacterial pathogens across the study years is likely due to the improved hygiene of food/water supplying in recent years, especially in rural areas[19]. Worryingly, increasing antibiotic usage has contributed to antibiotic resistance in recent years[20], which may be related to the increased detection of several bacterial pathogens. A better understanding of antimicrobial-resistant patterns of enteropathogens is an important avenue for future research.

We examined the viral pathogen interactions at the individual level by using results from patients who had all viruses tested. The detection of rotavirus was negatively correlated with all the other enteropathogens, which might be explained by the mechanism of competitive replication among coinfected viruses[21]. This finding can also be corroborated by previous research showing that rotavirus replication is susceptible to interference by other enteric viruses in the gut[22]. Norovirus was positively correlated with adenovirus and astrovirus, possibly owing to their shared transmission routes or susceptible population. The bacteria–bacteria and bacteria–virus interactions were not examined owing to low bacterial positive rate. The current knowledge on whether the circulation of one pathogen enhances or diminishes the infection incidence of another might lend to an enhanced estimation of the diagnosis or treatment choice. The elaboration of such interactions may have economic implications through public health planning, and the clinical management of diarrhea disease.

The simultaneous detection of a wide range of pathogens offered the opportunity of implementing a discrimination analysis between bacterial and viral diarrhea cases, which according to our data, could be attained by using age, presence of vomiting, and mucous stool. These findings may help refine clinical decision criteria for the selection of laboratory analyses to be performed on clinical samples. The degree to which the cost and benefits of such an approach might reduce hospital costs would also be a promising area of future research.

By analyzing aggregated nationwide data over ten years and among regions with wide meteorological and socio-economical variation, we can estimate these covariate influences on the prevalence of enteropathogens. In agreement with previous studies[23,24], we found most of the viral infections increased as temperature and humidity decreased and vice versa. These associations are supported by laboratory findings indicating that lower temperature and humidity increase the survival of viral enteropathogens in environment[25]. We found high wind speed, precipitation, and the sunshine hour was associated with reduced viral detection. Considering that infectious virus persists not only on surfaces but also in aerosolization of virus-laden dust particles[26], it is logical to deduce that viral survival in the air can be reduced by these factors, thus resulting in the decreased detection. These findings are also in agreement with previous modeling results that rotavirus transmission is enhanced when aerosols are able to linger in slow-moving air and inhibited by stronger winds that transport particles away from susceptible individuals[27].

Our study suggests that temperature is associated with the level of bacterial diarrhea infections, but this effect is less robust than those of population density and GDP, suggesting that crowded conditions and lower living standards disproportionately favor the infection. These findings are in accordance with existing evidence that the patients living in LMICs' areas were more susceptible to *Shigella* and other bacterial diarrhea, primarily due to poor access to healthcare, safe water, and sanitation, and low-income or marginalized populations[28,29].

We show differences in etiological structure among regions in China, revealing spatial variation in seasonal activity and dynamics. The climatic and socioeconomic factors that underlie these regional differences were investigated, based on which, we presented a novel pathogen clustering at the population scale. This information could assist in understanding the timing of pathogen activity at both national and subnational levels, which is important to healthcare providers and health officials who use this data to guide diagnostic testing, conduct disease surveillance, and response to outbreaks. In comparison with previous studies that were performed on limited enteropathogens, within narrow geographic regions, or with small-scale population[30–33] the current finding, especially regarding the influence of environmental factors, might have wider application than China.

Unexpectedly, we found a significantly longer time between onset and hospital admission for patients infected by a known enteropathogens than those tested negative for enteropathogens. We postulate that the longer delay between onset and hospital admission for pediatric patients than the adults, together with a significantly higher positive rate among the pediatric patients than the other age groups, contributes to this indirect relationship.

There are some limitations to this study. First, not all the listed pathogens were tested in all recruited patients as the reference laboratories had different test capacity and priorities of research interest, however, with the large number of patients that received comparable panels of testing pathogens (in absolute terms and across space and time), we are confident that the pathogen spectrum identified was not biased from that of either population. Second, the number of sentinel hospitals from Western China was less than those in the densely populated regions, as there was a lower than average population size in this region. This limitation may have hindered a full understanding of the dynamic pattern of the diarrhea pathogens in this region. Third, there were 73.93% of the patients who were negative for all of the 23 currently tested pathogens, who might be infected with organisms not included in our diagnostic tests and algorithms. In recent years, a growing list of emerging pathogens, especially viral pathogens were found to cause diarrhea, owing to the breakthroughs in the metagenomics field, such as Saffold cardiovirus[34], coronaviruses[35], picobirnaviruses[36], and MW polyomavirus[37] that have been proposed as "new" viral etiologic agents of diarrhea. The new-generation sequencing techniques which are of wide utility for all pathogens, though not yet adapted for widespread use in pathogen surveillance, should be strengthened for their application to identify novel pathogens to close the diagnostic gap.

In summary, our data provide a comprehensive understanding of the etiologic diagnosis of diarrhea, as well as their

demographic, seasonal, and geographic patterns in China. This allows an enhanced identification of predominant diarrheal pathogens candidates for diagnosis in clinical practice and targeted prevention control from the public health perspective at both national and sub-national levels. Continuous longitudinal surveillance is encouraged in order to maintain the insights to date and form a baseline for future epidemiological studies on diarrheal pathogens in China.

## Methods

**The active surveillance system.** Between January 2009 and December 2018, active surveillance of patients with acute diarrhea was administered in 217 sentinel hospitals and 93 reference laboratories in all 31 provinces (autonomous regions or municipalities) on the Chinese mainland which was managed by the Chinese Center for Diseases Control and Prevention (China CDC) (Supplementary Fig. 6). The numbers of sentinel hospitals and reference laboratories were determined in proportion to the total population size within each ecological region and the sentinel hospitals were chosen after careful consideration for the capacities of surveillance and laboratory testing and the representativeness of geographical locations (Supplementary Methods). All participating hospitals and laboratories used a surveillance protocol that included guidelines for patient enrollment, specimen collection, laboratory testing, data management, and other related standard operating procedures that were developed by China CDC[38]. A case of acute diarrhea was defined as the presence of ≥3 passages of watery, loose, mucus-, or bloody-stools within a 24-h period. Patients referred from other hospitals or patients not initially diagnosed in sentinel hospitals or patients with the non-infectious disease were excluded from this study[7].

**Specimen collection and laboratory tests.** For all participating patients, stool specimens were collected immediately after they were admitted into the hospital and before therapy was administered. For virological and parasitological testing, the stool was collected in sterilized containers without preservatives and tested as soon as possible, and if not, were stored at −80 °C until tested. For bacteriological testing, stool specimens were collected using five sterilized cotton swabs and immediately plated onto culture medium, and if not, were placed in Cary Blair Medium at 4 °C for transporting to the laboratory.

Seven viral pathogens were tested from the stool specimens. Rotavirus A antigen was tested by enzyme-linked immunosorbent assay, and G and P genotyping was performed by reverse transcription-polymerase chain reaction (RT-PCR) (Supplementary Table 12-1). Testing of norovirus, adenovirus, astrovirus, sapovirus, rotavirus B, and rotavirus C was performed by polymerase chain reaction (PCR) or RT-PCR (Supplementary Table 12-2). Thirteen bacterial pathogens were tested by performing isolation with or without enrichment procedures at the first step. For *Y. enterocolitica*, *Y. pseudotuberculosis*, DEC, *C. jejuni* and *C. coli*, the isolation was subsequently tested by PCR (Supplementary Table 12-3,4,5), and for NTS, *V. parahaemolyticus*, *V. cholerae*, *Vibrio fluvialis* (*V. fluvialis*), *Vibrio mimicus* (*V. mimicus*), *A. hydrophila*, *P. shigelloides* and *Shigella*, the isolation was subsequently tested by biochemical and serological assays. Altogether three parasites, including *E. histolytica*, *G. lamblia,* and *Cryptosporidium* were tested by direct microscopy, commercial immunoassay, and PCR (Supplementary Table 12-6) as guided (Supplementary Fig. 7).

The National Health Commission of the People's Republic of China decided that since data from patients with diarrhea was part of continuing public health surveillance and implemented national surveillance guidelines; parents/guardians of participants in this study were only required to provide brief verbal consent during their enrollment, which was recorded in each questionnaire by their physicians. This project and the above procedure for obtaining consent were approved by the ethical review committee of China CDC (2015-025).

**Statistical analysis.** Individual data about demography, clinical manifestations, laboratory testing results, medication use, and outcomes were collected by reviewing medical records and the data were entered into a standardized database by trained clinicians. All the data were uploaded to the online management system structured by the China CDC, sorted to remove redundant data, and checked for incomplete records. The positive rate of any specific pathogen was calculated by dividing the number of positive specimens by the total number tested for that pathogen. Descriptive statistics were performed for all variables. The pathogen interactions at the individual host level were tested by multivariable binary logistic regression. Holm's method was applied to control the probability of one or more false discoveries arising[39]. We used the binary eXtreme Gradient Boosting (Supplementary Methods) to differentiate viral infections and bacterial infections. We reviewed the literature to identify all meteorological variables that had a significant effect on the diarrheal enteropathogens, which were used for the current analysis[23,24,40]. The associations between daily positive detection and six meteorological indicators (including daily average temperature, daily average relative humidity, daily sunshine duration, daily precipitation, daily average wind speed, and daily average air pressure) and four sociological factors (GDP per capita,

population density, proportion of children, and proportion of the older people) were explored by using a negative binomial regression at the sentinel city level. Univariate analysis was performed to examine the effect of each variable. Multivariate analysis was then performed using variables with a *p*-value < 0.2 from the univariate analysis as covariates and the final model was reached by excluding variables with *p*-values of > 0.05. Spearman correlation tests were conducted to evaluate the correlations between covariates, and collinearity between variables was avoided by removing the one with less significance from the model. Models were optimized by comparing Akaike information criterion values when correlated variables were added or removed.

Statistical analysis was performed using R statistical software (version 3.5.3), SAS software version 9.4, ArcGIS software version 10.5, and Joinpoint Trend Analysis Software (version 4.7.0.0) (Statistical Research and Applications Branch, National Cancer Institute, USA) (Supplementary Methods). All statistical tests were two-sided and *p* < 0.05 were statistically significant.

**Reporting summary.** Further information on research design is available in the Nature Research Reporting Summary linked to this article.

## Data availability

Relevant data that support the findings of this study and model results generated as part of this study are publicly available within the paper and its Supplementary Information Files. Raw data are not publicly available due to restrictions by the data provider, which were used under license for the current study, but are available upon reasonable request to the corresponding author and with permission from the data provider (Li-Ping Wang). Source data are provided with the paper. Source data are provided with this paper.

## Code availability

The code that supports all findings of this study is available from the corresponding authors upon reasonable request.

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

## Acknowledgements
We would like to thank all the subjects, their families, and collaborating clinicians for their participation. This work was financially supported by grants from the China Mega-Project on Infectious Disease Prevention (Nos. 2018ZX10713001, 2018ZX10713002, 2018ZX10201001, and 2017ZX10103004), the National Natural Science Funds (Nos. 91846302, 81825019). The funders had no role in the design and conduct of the study; collection, management, analysis, and interpretation of the data; preparation, review, or approval of the manuscript; and decision to submit the manuscript for publication. We thank staff members of the Department of Science and Education of the National Health Commission, and the diarrheal surveillance network laboratories and sentinel hospitals in the participating 31 provinces of China for assistance with field investigation, administration, and data collection.

## Author contributions
L.P.W., Z.J.L., W.Z.Y., H.Q.J., G.F.G., L.Q.F., and W.L. conceived, designed, and supervised the study. X.W., L.S.S., X.R., Y.F.W., S.H.L., C.H.Z., M.J.G., X.A.Z., J.L., S.W.Z., Z.G.Y., X.C., Z.S.Y., L.M., X.H.W., Y.L.L., A.L.C., S.J.L., and Y.L.Z. collected data. S.X.Z., Q.B.L., L.S.S., X.R., H.Y.Z., X.A.Z., M.Y.L., and Y.L.Z. cleaned data. S.X.Z., Q.B.L., H.Y.Z., L.S.S., and Y.L.Z. analyzed the data supervised by L.P.W., L.Q.F., and W.L. W.L., S.X.Z., L.P.W., Z.J.L., and L.Q.F. wrote the drafts of the manuscript. L.P.W., W.L., L.Q.F., Z.J.L., X.W., W.B.X, Y.C., J.G.W., Z.H.Y., M.F.L., and L.Y.H. interpreted the findings. Z.J.L., L.P.W., L.Q.F., W.L., and S.I.H commented on and revised drafts of the manuscript. All authors read and approved the final report.

## Competing interests
The authors declare no competing interests.

## Additional information

## The Chinese Centers for Disease Control and Prevention (CDC) Etiology of Diarrhea Surveillance Study Team

Wei-Zhong Yang[21], George F. Gao[21], Zhong-Jie Li[22], Li-Ping Wang[22], Xiang Ren[22], Yi-Fei Wang[22], Sheng-Hong Lin[22], Cui-Hong Zhang[22], Meng-Jie Geng[22], Xin Wang[23], Huai-Qi Jing[23], Wen-Bo Xu[24], Ai-Li Cui[24], Yu-Juan Shen[25], Yan-Yan Jiang[25], Qiao Sun[26], Li-Peng Hao[26], Chu-Chu Ye[26], Wei Liu[27], Xiao-Ai Zhang[27], Liu-Yu Huang[28], Yong Wang[28], Wen-Yi Zhang[28], Ying-Le Liu[29], Jian-Guo Wu[29], Qi Zhang[29], Wei-Yong Liu[30], Zi-Yong Sun[30], Fa-Xian Zhan[31], Ying Xiong[32], Lei Meng[33], De-Shan Yu[33], Chun-Xiang Wang[34], Sheng-Cang Zhao[34], Wen-Rui Wang[35], Xia Lei[35], Juan-Sheng Li[36], Yu-Hong Wang[37], Yan Zhang[37], Jun-Peng Yang[38], Yan-Bo Wang[38], Fu-Cai Quan[39], Zhi-Jun Xiong[39], Li-Ping Liang[40], Quan-E Chang[40], Yun Wang[41], Ping Wang[41], Zuo-Sen Yang[42], Ling-Ling Mao[42], Jia-Meng Li[43], Li-Kun Lv[43], Jun Xu[44], Chang Shu[44], Xiao Chen[45], Yu Chen[45], Yan-Jun Zhang[46], Lun-Biao Cui[47], Kui-Cheng Zheng[48], Xing-Guo Zhang[49], Xi Zhang[50], Li-Hong Tu[50], Zhi-Gang Yi[51], Wei Wang[51], Shi-Wen Zhao[52], Xiao-Fang Zhou[52], Xiao-Fang Pei[53], Tian-Li Zheng[53], Xiao-Ni Zhong[54], Qin Li[55], Hua Ling[55], Ding-Ming Wang[56], Shi-Jun Li[56], Shu-Sen He[57], Meng-Feng Li[58], Jun Li[58], Xun Zhu[58], Chang-Wen Ke[59], Hong Xiao[59], Biao Di[60], Ying Zhang[60], Hong-Wei Zhou[61], Nan Yu[61], Hong-Jian Li[62], Fang Yang[62], Fu-Xiang Wang[63] & Jun Wang[63]

[21]Chinese Center for Disease Control and Prevention, Beijing, China. [22]Division of Infectious Disease, Key Laboratory of Surveillance and Early-warning on Infectious Disease, Chinese Center for Disease Control and Prevention, Beijing, China. [23]National Institute for Communicable Disease Control and Prevention, Chinese Center for Disease Control and Prevention, Beijing, China. [24]National Institute for Viral Disease Control and Prevention, Chinese Center for Disease Control and Prevention, Beijing, China. [25]National Institute of Parasitic Diseases, Chinese Center for Disease Control and Prevention, Shanghai, China. [26]Center of Disease Prevention and Control in Pudong New Area of Shanghai, Shanghai, China. [27]State Key Laboratory of Pathogen and Biosecurity, Beijing Institute of Microbiology and Epidemiology, Beijing, China. [28]The Institute for Disease Prevention and Control of PLA, Beijing, China. [29]Wuhan University, Wuhan, China. [30]Tongji Hospital, Tongji Medical College, Huazhong University of Science and Technology, Wuhan, China. [31]Hubei Provincial Center for Disease Control and Prevention, Wuhan, China. [32]Jiangxi Provincial Center for Disease Control and Prevention, Nanchang, China. [33]Gansu Provincial Center for Disease Control and Prevention, Lanzhou, China. [34]Qinghai Provincial Center for Disease Control and Prevention, Xining, China. [35]Inner Mongolia Autonomous Region Comprehensive Center for Disease Control and Prevention, Hohhot, China. [36]Lanzhou University, Lanzhou, China. [37]Lanzhou Center for Disease Control and Prevention, Lanzhou, China. [38]Baiyin Center for Disease Control and Prevention, Baiyin, China. [39]Tianshui Center for Disease Control and Prevention, Tianshui, China. [40]Wuwei Center for Disease Prevention and Control, Wuwei, China. [41]Qingyang Center for Disease Control and Prevention, Qingyang, China. [42]Liaoning Provincial Center for Disease Control and Prevention, Shenyang, China. [43]Tianjin Center for Disease Control and Prevention, Tianjin, China. [44]Heilongjiang Provincial Center for Disease Control and Prevention, Harbin, China. [45]Zhejiang University, Hangzhou, China. [46]Zhejiang Center for Disease Control and Prevention, Hangzhou, China. [47]Jiangsu Provincial Center for Disease Control and Prevention, Nanjing, China. [48]Fujian Center for Disease Control and Prevention, Fuzhou, China. [49]Beilun People's Hospital, Ningbo, China. [50]Shanghai Municipal Center for Disease Control and Prevention, Shanghai, China. [51]Shanghai Public Health Clinical Center, Shanghai, China. [52]Yunnan Center for Disease Control and Prevention, Kunming, China. [53]Sichuan University, Chengdu, China. [54]Chongqing Medical University, Chongqing, China. [55]Chongqing Center for Disease Control and Prevention, Chongqing, China. [56]Guizhou Center for Disease Control and Prevention, Guiyang, China. [57]Sichuan Province Center for Disease Control and Prevention, Chengdu, China. [58]Sun Yat-sen University, Guangzhou, China. [59]Guangdong Provincial Center for Disease Control and Prevention, Guangzhou, China. [60]Guangzhou Municipal Center for Disease Control and Prevention, Guangzhou, China. [61]Zhujiang Hospital, Southern Medical University, Guangzhou, China. [62]Jinan University, Guangzhou, China. [63]The Third People's Hospital of Shenzhen, Shenzhen, China.

