## [Peer Review File · Nature Communications]

REVIEWER COMMENTS

Reviewer #1 (Remarks to the Author):

The study "Etiological, epidemiological, and clinical features of acute diarrhea in China, 2009-2018: an active sentinel surveillance study" is a long-term, comprehensive analysis of the pathogens that cause gastroenteritis in all age groups of the population covering widespread regions in China.

Between 2009 and 2018, a national sentinel surveillance programme enrolled 157 792 patients at 217 sentinel hospitals in 31 provinces of China. A total of 152 792 patient met the inclusion criteria. The participating laboratories screened for seven viruses, 13 bacteria and three parasites. A vast amount of data is presented in the paper, evidenced by the 17 supplementary tables and figures that accompany the paper.

The authors detected rotavirus as the predominant pathogen overall in the study (20.4%) followed by norovirus (12.47%), diarrheagenic E. coli (6.71%) and nontyphoidal Salmonella (4.41%), adenovirus (3.33%), with the various other pathogens below 3%. Children under 5 years of age tested positive for viral pathogens more frequently, whereas bacterial pathogens were detected more commonly in adults (18-45 years). Interesting joinpoint regression analyses indicated that the detection rate of rotavirus, norovirus and adenovirus peaked at 2 – 3 years of age and that norovirus displayed a second detection peak at 16 years of age. Rotavirus was the leading pathogen in children and norovirus the leading pathogen in adults. Compared to all other pathogens norovirus detection rates remained high across the adult age groups. Overall rotavirus A G9P[8] dominated followed by G3P[8] and G1P[8]. Norovirus GII was predominant, no strain genotype information was provided for norovirus. China was subdivided into seven ecological regions and seasonal patterns were investigated at the sentinel city level with various meteorological and four sociological factors. Virus infections exhibited winter/spring seasonality and bacterial infections summer/autumn seasonality. The seasonal patterns were less pronounced in the subtropical and tropical regions. The proportion of children in a sentinel city affected virus infections, whereas population density significantly affected bacterial infections.

This is a very informative study, the results are presented in a clear and comprehensive manner. Statistical analysis is well described and appropriate. The methodology is described adequately. The paper is well written. There are however some suggested corrections and a few clarifications are required.

Major comments

1) The total number of included participants was 152 792. All of these patients were not tested for each of the 23 gastroenteritis pathogens. This is understandable due to the size of the study and the number of sites involved. However, it should be stated clearly at the start of the results section to avoid confusion. I suggest that Supplementary Figure 3 be moved to the manuscript to provide a clear overview of the number specimens that were tested for each type of pathogen. The authors might expand on the figure to include the numbers tested for each pathogen. The authors should also move the "Results" section in Supplementary data on page 3/24 to the Results section of the main manuscript, to avoid confusion.

2) Another concern is that the authors stated a number of 28,811/85,731 patients who had at least one viral pathogen detected. Since all 85,731 patients were not tested for all the viruses, I do not think this is a very meaningful number, since it could be much higher if all 85,731 specimens were tested. I think it would be more informative if the authors present the numbers of the group of specimens that were tested for all viruses or bacteria, before they present the pathogen specific data.

3) Do the authors have access to data on the length of hospitalization and mortality associated with each admission? In the "Statistical analysis" section on page 8 they state that "outcomes

were collected by reviewing medical records". But no mention is made of the number of deaths recorded in the study population. The authors did include data on the time that elapsed between onset of symptoms to hospitalization, but the length of hospital stay and eventual outcome is also of interest. This could be added to Supplementary Table 9, if available.

4) The authors should state the proportion of cases from urban and from rural settings in the section that describes the study population in the results.

Minor corrections:

Page 4, Line 52: ...in Chinese mainland.....change to...on the Chinese mainland

Page 4, Line 63: ...had affected on most...change to.....had effects on most....

Page 4, Line 70: ...how enteropathogens change in diarrheal patients.....change to.....how the distribution of enteropathogens change in diarrheal patients.....

Page 5, Line 96:detection assay among studies.....change to detection assays among studies

Page 5, Line 96/97: network for the patients.....change to network for patients..

Page 7, Line 141:disclosed similar trend....change to....disclosed a similar trend..

Page 7, line 152 and 153: DEC and NTS are used here without definition. The abbreviations are defined in the Methods section, but since that is only at the end of the manuscript, it should rather be defined here at first use.

Page 10, Line 239: Please rephrase the first sentence in line 239 to improve clarity.

Page 11, Line 244: patter.....change to....pattern

Page 12, Line 252:Age effect on the pathogen.....change to.....The effect of age on the pathogen....

Page 12, Line 253: ...trend for patient with younger age....change to....trend for younger patients, especially < 5 years old

Page 12, Line 254: ...and higher variety....change to....and a higher variety...

Page 13, Line 275: ..that related to age....change to.....that is related to age

Page 13, Line 277: Consider rephrasing as follows:

...norovirus diagnosis and prevention, once a vaccine is available, should be a priority in the 2-3 year age group, and additionally at the age of five years.

Page 14, Line 327: ...not only on surface....change to....not only on surfaces..

Page 17, Line 385: ...the presence >3 passages....change to.....the presence of >3 passages

Page 18, Line 425:reviewed literatures.....change to.....reviewed the literature.....

Page 18, Line 426: ...had significant effect....change to....had a significant effect

Page 18 – Please provide the primer/probe sequences that were used to the detection PCRs/RT-PCRs in the supplementary section.

Supplementary appendix

Page 3:

At the start of the results paragraphs use "In total" instead of "Totally"

Page 4, Line 4: ...each age groups.....change to....each age group.....

Supplementary Figure 2:

A – Norovirus is not listed as a virus tested for in the block PCR or RT-PCR.....etc....

B - Carry to the microbiology laboratory....change to.....Transported to the microbiology laboratory....

Supplementary Table 2: Please provide the time unit for delay. I assume it is days from onset to admission, but it should be stated for clarity.

Reviewer #2 (Remarks to the Author):

The manuscript presents findings from surveillance efforts across 217 hospitals to understand the characteristics of clinically attended diarrhoeal disease in China over 10 years. To my knowledge, analyses of diarrhoeal disease surveillance at this geographical and time scale is unprecedented. Diarrhoeal disease remains a global health concern and this work provides a valuable, thorough description of the enteric pathogens and some of the environmental factors that might influence the diarrhoeal disease burden in China.

The analysis focuses on the proportion of samples testing positive for specific enteric pathogens. It would be interesting to understand how this proportion relates to trends in diarrhoeal disease incidence (i.e. absolute number of cases captured by the surveillance system over time and by region).

The discussion could better describe how findings from this work align or contrast with previous, smaller-scale studies seeking to identify enteric pathogens responsible for the diarrhoeal disease burden in other contexts (eg. GEMS study; Lambisia et al., 2020; ...) and whether some of these findings, especially regarding the influence of environmental factors, might be applicable outside China. Clearer recommendations for future research – and diarrhoeal disease surveillance – could also be formulated.

The manuscript would benefit from grammar and spelling checks. I have noted in comments likely confusions between “tested” and “detected” that alter the meaning of the text and tables.

Other minor comments are noted below.

Lines 79-82: an estimated 1.6 million deaths per year (time period missing).

Line 111-112: Sup. Fig 4 suggests a decreasing trend in the number of patients enrolled each year; was this uniform across regions? and pathogens?

Lines 112-114: it is unclear why the denominator here is so small (3,330); is that a yearly average?

Lines 114-115: the finding of a significantly longer time between onset and hospital admission for patients infected by a known enteropathogens is not commented in Discussion. Do you have any hypothesis regarding why such a difference was observed?

Lines 118-119, 151-152, 178-179: the likelihood of detecting at least one pathogen depends on the number of pathogens tested for. How was this taken into account given that not all samples appear to have undergone the same tests?

Lines 135-138: include denominators for all %.

Lines 145-147: this sentence is unclear.

Lines 152, 159: abbreviations should be defined upon first occurrence for clarity.

Line 189: how were viral, bacterial and parasitic “co-infection rates” defined? Infection with two or more viruses, bacteria, parasites?

Line 265: suggest to rephrase – this could be attributed to ...

Lines 289-290: 20% of all child deaths over which time period?

Line 297: if data available on water supply and sanitation services coverage over the study period, it would be an interesting aspect to consider in the analyses, in addition to the rural/urban categories.

Lines 297-300: better understanding AMR patterns (not only among bacteria) is an important avenue for future research.

Lines 309-313: it is unclear to me what the authors are suggesting here, concretely.

Lines 314-319: do the authors feel that their dataset is sufficient to propose a diagnostic tree to distinguish between bacterial, viral and parasitic infections among Chinese patients with acute diarrhoea? Otherwise, would using findings from their study to refine decision criteria for the selection of laboratory analyses to be performed on clinical samples be a more realistic recommendation?

Line 349: detected or tested?

A summary of the decision criteria used to decide which pathogens to test for in a given sample and whether they might have led to bias in the detection of certain pathogens would be helpful. Reporting the frequency of testing for each pathogen could also address this point.

Supplementary Materials and Methods:

- Ref. 4 – the URL doesn't seem to be accessible. How was the rural/urban status defined for a given patient address?

Sup. Fig 3:

- related to my previous comment, were samples tested for parasites also tested for viruses and bacteria in some cases? If not, why?
- The figure appears in contradiction with the statement on page 2 of Supplementary Methods and Materials: "Parasites are not included in the study of pathogen co-infection patterns..."
- Bacterium (singular) and bacteria (plural) should be inverted on this figure.

Sup. Table 1:

- headings are unclear – what does "detect" mean here? Should it be replaced with "tested", as in Fig 3?

Suggest to include information on rural/urban categories in Supplementary Tables 1, 2, 3 if available.

Sup. Table 4: it seems like the line headings like "All viruses detected" should be replaced with "All viruses tested" (same for viruses, parasites).

Responses to the reviewers:

Reviewer #1 (Remarks to the Author):

The study “Etiological, epidemiological, and clinical features of acute diarrhea in China, 2009-2018: an active sentinel surveillance study” is a long-term, comprehensive analysis of the pathogens that cause gastroenteritis in all age groups of the population covering widespread regions in China.

Between 2009 and 2018, a national sentinel surveillance programme enrolled 157 883 patients at 217 sentinel hospitals in 31 provinces of China. A total of 152 792 patient met the inclusion criteria. The participating laboratories screened for seven viruses, 13 bacteria and three parasites. A vast amount of data is presented in the paper, evidenced by the 17 supplementary tables and figures that accompany the paper.

The authors detected rotavirus as the predominant pathogen overall in the study (20.4%) followed by norovirus (12.47%), diarrheagenic E. coli (6.71%) and nontyphoidal Salmonella (4.41%), adenovirus (3.33%), with the various other pathogens below 3%. Children under 5 years of age tested positive for viral pathogens more frequently, whereas bacterial pathogens were detected more commonly in adults (18-45 years). Interesting joinpoint regression analyses indicated that the detection rate of rotavirus, norovirus and adenovirus peaked at 2 – 3 years of age and that norovirus displayed a second detection peak at 16 years of age. Rotavirus was the leading pathogen in children and norovirus the leading pathogen in adults. Compared to all other pathogens norovirus detection rates remained high across the adult age groups. Overall rotavirus A G9P[8] dominated followed by G3P[8] and G1P[8]. Norovirus GII was predominant, no strain genotype information was provided for norovirus. China was subdivided into seven ecological regions and seasonal patterns were investigated at the sentinel city level with various meteorological and four sociological factors. Virus infections exhibited winter/spring seasonality and bacterial infections summer/autumn seasonality. The seasonal patterns were less pronounced in the subtropical and tropical regions. The proportion of children in a sentinel city affected virus infections, whereas population density significantly affected bacterial infections.

This is a very informative study, the results are presented in a clear and comprehensive manner. Statistical analysis is well described and appropriate. The methodology is described adequately. The paper is well written. There are however some suggested corrections and a few clarifications are required.

[Response] We appreciate the reviewer’s summary, positive tone and helpful comments that we believe have helped us better communicate our findings.

Major comments

1) The total number of included participants was 152 792. All of these patients were not tested for each of the 23 gastroenteritis pathogens. This is understandable due to

the size of the study and the number of sites involved. However, it should be stated clearly at the start of the results section to avoid confusion. I suggest that Supplementary Figure 3 be moved to the manuscript to provide a clear overview of the number specimens that were tested for each type of pathogen. The authors might expand on the figure to include the numbers tested for each pathogen. The authors should also move the “Results” section in Supplementary data on page 3/24 to the Results section of the main manuscript, to avoid confusion.

[Response] We appreciate the reviewer’s valuable comments. As suggested, we have stated the tested number of pathogens clearly at the start of the results section (Page 6, Lines 109–113). We have moved “Supplementary Fig. 3” to main text “Fig. 1” and expanded on the figure to include the numbers tested for each pathogen (Page 30). We have moved the “Supplementary Results” section to the main text “Results” in the revised manuscript (Pages 6–8, Lines 125–161).

2) Another concern is that the authors stated a number of 28,811/85,731 patients who had at least one viral pathogen detected. Since all 85,731 patients were not tested for all the viruses, I do not think this is a very meaningful number, since it could be much higher if all 85,731 specimens were tested. I think it would be more informative if the authors present the numbers of the group of specimens that were tested for all viruses or bacteria, before they present the pathogen specific data.

[Response] We appreciate the reviewer’s comments. We agree with the reviewer that the sample groups tested for all viruses or bacteria provided more informative data. As suggested, we have moved the “Supplementary Results” section on Page 3 of the Supplementary Information to the Results section of the main text, ahead of the pathogen specific data in the revised manuscript (Pages 6–8, Lines 125–161).

3) Do the authors have access to data on the length of hospitalization and mortality associated with each admission? In the “Statistical analysis” section on page 8 they state that “outcomes were collected by reviewing medical records”. But no mention is made of the number of deaths recorded in the study population. The authors did include data on the time that elapsed between onset of symptoms to hospitalization, but the length of hospital stay and eventual outcome is also of interest. This could be added to Supplementary Table 9, if available.

[Response] We appreciate the reviewer’s comments. According to our data, 89,441 patients had a definite outcome recorded, among whom 40 patients (0.04%) died. We have described the outcome data in the revised manuscript (Page 6, Lines 117–118) and supplemented the outcome data in Supplementary Table 1 (Page 7 in the Supplementary Information). We agree that the length of hospitalization would be a useful indicators of disease outcome, but unfortunately, these were not available in this study.

4) The authors should state the proportion of cases from urban and from rural settings in the section that describes the study population in the results.

[Response] We appreciate the reviewer’s comments. We have supplemented data on

the proportion of cases from urban and from rural settings to the main text (Page 6, Lines 116–117) and Supplementary Table 1 (Page 7 in the Supplementary Information) of the revised manuscript.

Minor corrections:

Page 4, Line 52: ...in Chinese mainland.....change to...on the Chinese mainland
[Response] Revised as suggested (Page 4, Line 52).

Page 4, Line 63: ...had affected on most...change to.....had effects on most....
[Response] Revised as suggested (Page 4, Line 63).

Page 4, Line 70:...how enteropathogens change in diarrheal patients.....change to.....how the distribution of enteropathogens change in diarrheal patients.....
[Response] Revised as suggested (Page 4, Line 70).

Page 5, Line 96:detection assay among studies.....change to detection assays among studies
[Response] Revised as suggested (Page 5, Line 97).

Page 5, Line 96/97: network for the patients.....change to network for patients..
[Response] Revised as suggested (Page 5, Line 98).

Page 7, Line 141:disclosed similar trend....change to....disclosed a similar trend...
[Response] Revised as suggested (Page 8, Line 186).

Page 7, line 152 and 153: DEC and NTS are used here without definition. The abbreviations are defined in the Methods section, but since that is only at the end of the manuscript, it should rather be defined here at first use.
[Response] We appreciate the reviewer's corrections. We have defined all abbreviations at the first time they appear.

Page 10, Line 239: Please rephrase the first sentence in line 239 to improve clarity.
[Response] We appreciate the reviewer's comments. We have rephrased the first sentence into "The seasonal patterns of enteropathogen also differed regarding the geographical locations." (Page 13, Lines 303–304).

Page 11, Line 244: patter.....change to....pattern
[Response] Revised as suggested (Page 13, Line 308).

Page 12, Line 252:Age effect on the pathogen.....change to.....The effect of age on the pathogen....
[Response] Revised as suggested (Page 14, Line 316).

Page 12, Line 253: ...trend for patient with younger age....change to....trend for younger patients, especially < 5 years old
[Response] Revised as suggested (Page 14, Line 317).

Page 12, Line 254: ...and higher variety....change to....and a higher variety...
[Response] Revised as suggested (Page 14, Line 318).

Page 13, Line 275: ..that related to age....change to.....that is related to age
[Response] Revised as suggested (Page 15, Line 342).

Page 13, Line 277: Consider rephrasing as follows:
...norovirus diagnosis and prevention, once a vaccine is available, should be a priority in the 2-3 year age group, and additionally at the age of five years.
[Response] Revised as suggested (Page 15, Lines 343–345).

Page 14, Line 327: ...not only on surface....change to....not only on surfaces...
[Response] Revised as suggested (Page 17, Line 399).

Page 17, Line 385: ...the presence >3 passages....change to.....the presence of >3 passages
[Response] Revised as suggested (Page 20, Line 470).

Page 18, Line 425:reviewed literatures.....change to.....reviewed the literature.....
[Response] Revised as suggested (Page 22, Line 516).

Page 18, Line 426: ...had significant effect....change to....had a significant effect
[Response] Revised as suggested (Page 22, Line 517).

Page 18 – Please provide the primer/probe sequences that were used to the detection PCRs/RT-PCRs in the supplementary section.
[Response] We appreciate the reviewer’s comments. As suggested, we have provided the primer/probe sequences that were used to the detection PCRs/RT-PCRs in the Supplementary Table 13 (Pages 28–33 in the Supplementary Information).

Supplementary appendix

Page 3:

At the start of the results paragraphs use “In total” instead of ”Totally”
[Response] Revised as suggested (Page 6, Line 126; Page 7, Line 138; Page 7, Line 156).

Page 4, Line 4: ...each age groups.....change to....each age group.....
[Response] We appreciate the reviewer’s comments. We have rephrased the sentence into “Overall, DEC and NTS were the leading bacterial pathogens that were identified from pediatric patients <18 years old.” (Page 7, Lines 145–147).

Supplementary Figure 2:

A – Norovirus is not listed as a virus tested for in the block PCR or RT-PCR.....etc....

B - Carry to the microbiology laboratory....change to.....Transported to the microbiology laboratory....

[Response] We appreciate the reviewer's comments. As suggested, we have listed norovirus as a virus tested for in the block PCR or RT-PCR and modified "Carry to the microbiology laboratory" into "Transported to the microbiology laboratory" in Supplementary Fig. 7 (Page 34 in the Supplementary Information).

Supplementary Table 2: Please provide the time unit for delay. I assume it is days from onset to admission, but it should be stated for clarity.

[Response] We appreciate the reviewer's comments. Yes, the time unit for delay is day, which had been added to the Supplementary Tables 2, 3 and 10 (Page 9; Page 10; Page 21 in the Supplementary Information).

Reviewer #2 (Remarks to the Author):

The manuscript presents findings from surveillance efforts across 217 hospitals to understand the characteristics of clinically attended diarrhoeal disease in China over 10 years. To my knowledge, analyses of diarrhoeal disease surveillance at this geographical and time scale is unprecedented. Diarrhoeal disease remains a global health concern and this work provides a valuable, thorough description of the enteric pathogens and some of the environmental factors that might influence the diarrhoeal disease burden in China.

The analysis focuses on the proportion of samples testing positive for specific enteric pathogens. It would be interesting to understand how this proportion relates to trends in diarrhoeal disease incidence (i.e. absolute number of cases captured by the surveillance system over time and by region).

[Response] We appreciate the reviewer's comments. As suggested, we have displayed the absolute number of tested cases and the proportion of at least one pathogen for each of the study year in Supplementary Fig. 1 (Page 8 in the Supplementary Information), however, we found no relation between them.

We have also displayed the absolute number of tested cases and the proportion of at least one pathogen for each of the study regions in the figure as follows, again, we found no relation between them.

The discussion could better describe how findings from this work align or contrast with previous, smaller-scale studies seeking to identify enteric pathogens responsible for the diarrhoeal disease burden in other contexts (eg. GEMS study; Lambisia et al., 2020; ...) and whether some of these findings, especially regarding the influence of environmental factors, might be applicable outside China. Clearer recommendations for future research – and diarrhoeal disease surveillance – could also be formulated.

[Response] We appreciate the reviewer’s helpful comments. We have compared the current findings with previous similar studies in the discussion:

“In this study, rotavirus was found to be the predominant pathogen overall, also the leading viral pathogen among children, while norovirus was the leading pathogen in adults. This is consistent with the previous findings¹⁰” (Page 14, Lines 327–330),

“We found a slow decrease in the detection of rotavirus across the study years, probably owing to the rotavirus vaccine interventions that had been advocated after the year of 2000¹⁴. This was in also line with previous results showing a similar decreased rate of rotavirus during post-rotavirus vaccine introduction in Coastal Kenya¹⁵.” (Page 15, Lines 346–350);

We have also discussed the potential application of the current findings outside China. “In comparison with previous similar studies that were performed on limited enteropathogens, within narrow geographic regions, or with small-scale population³⁰⁻³³ the current finding, especially regarding the influence of environmental factors, might have wider application than China.” (Pages 17–18, Lines 420–423 in the revised manuscript)

10. Kotloff, K. L. et al. The incidence, aetiology, and adverse clinical consequences of less severe diarrhoeal episodes among infants and children residing in low-income and middle-income countries: a 12-month case-control study as a follow-on to the Global Enteric Multicenter Study (GEMS). *Lancet Glob Health* 7, e568-e584 (2019).

15. Lambisia, A. W. et al. Epidemiological trends of five common diarrhea-associated enteric viruses pre- and post-rotavirus vaccine introduction in Coastal Kenya. *Pathogens (Basel, Switzerland)* 9, (2020).

30. Levy, K., Hubbard, A. E. & Eisenberg, J. N. Seasonality of rotavirus disease in the tropics: a systematic review and meta-analysis. *Int. J. Epidemiol.* 38, 1487-1496 (2009).

31. Ureña-Castro, K. et al. Seasonality of rotavirus hospitalizations at Costa Rica's national children's hospital in 2010-2015. *Int. J. Environ. Res. Public Health* 16, (2019).

32. Celik, C. et al. Rotavirus and adenovirus gastroenteritis: time series analysis. *Pediatr. Int.* 57, 590-596 (2015).

33. Sumi, A. et al. Effect of temperature, relative humidity and rainfall on rotavirus infections in Kolkata, India. *Epidemiol. Infect.* 141, 1652-1661 (2013).

Recommendations for future research were added to the last paragraph of the discussion “Continuous longitudinal surveillance is encouraged in order to maintain the insights to date and form a baseline for future epidemiological studies on diarrheal pathogens in China.” (Page 19, Lines 452–454).

The manuscript would benefit from grammar and spelling checks. I have noted in comments likely confusions between “tested” and “detected” that alter the meaning of the text and tables.

[Response] We appreciate the reviewer’s comments. We have used “tested” to represent the sample that had undergone the test assay, while used the detected positive to represent the positive results, which had been modified in the revised manuscript.

minor comments :

Lines 79-82: an estimated 1.6 million deaths per year (time period missing).

[Response] We appreciate the reviewer’s comments. We have supplemented the time period “in 2016” (Page 5, Line 82).

Line 111-112: Sup. Fig 4 suggests a decreasing trend in the number of patients enrolled each year; was this uniform across regions? and pathogens?

[Response] We appreciate the reviewer’s comments. We have provided the annual number of patients enrolled across seven ecological regions in the following figure, based on which, we found no consistently decreasing tend in the number of patients enrolled each year.

Lines 112-114: it is unclear why the denominator here is so small (3,330); is that a yearly average?

[Response] We appreciate the reviewer’s comments. These 3,330 patients referred to

those who have been tested for all the 23 gastroenteritis pathogens, including seven viral pathogens, thirteen bacterial pathogens and three parasites. Although 25,239 samples had been tested for both viral and bacterial pathogens, this number had been reduced to 3,330, due to small sample number (N=11,167) that were tested for parasites. We have discussed this limitation in the revised manuscript (Page 18, Lines 429–434).

Lines 114-115: the finding of a significantly longer time between onset and hospital admission for patients infected by a known enteropathogens is not commented in Discussion. Do you have any hypothesis regarding why such a difference was observed?

[Response] We appreciate the reviewer's valuable comments. We cannot confirm the reason underlying this difference for sure, but we indeed found a longer delay between onset and hospital admission among pediatric patients than among the adults (median delay 3 days in the Supplementary Table 2), while the pediatric patients also had a significantly higher positive rate than the other age groups (60.06%, 636/1059, in the Supplementary Table 2). This might lead to an indirect relationship between longer time between onset and hospital admission for patients infected by a known enteropathogens. We have commented this finding in the revised manuscript (Page 18, Lines 424–428).

Lines 118-119, 151-152, 178-179: the likelihood of detecting at least one pathogen depends on the number of pathogens tested for. How was this taken into account given that not all samples appear to have undergone the same tests?

[Response] We appreciate the reviewer's valuable comments. We agree with the reviewer that the number of tested pathogens determined the positive rate to a large extent. To report the rate of detecting at least one pathogen might be misleading under this situation, thus we have removed this rate for virus, bacterium, or parasite in the revised manuscript.

Lines 135-138: include denominators for all %.

[Response] Done as suggested.

Lines 145-147: this sentence is unclear.

[Response] We appreciate the reviewer's corrections and suggestions. We have rephrased the sentence and split it into two sentences. (Page 9, Lines 190–193).

Lines 152, 159: abbreviations should be defined upon first occurrence for clarity.

[Response] We appreciate the reviewer's corrections and suggestions. We have defined all abbreviations the first time they appear.

Line 189: how were viral, bacterial and parasitic “co-infection rates” defined? Infection with two or more viruses, bacteria, parasites?

[Response] We appreciate the reviewer's corrections and suggestions. For viruses, co-

infection rates mean co- infection rates among viruses. For bacteria, co- infection rates mean co- infection among bacteria. For parasites, co- infection rates mean co- infection among parasites. We have modified this expression in the results section of the revised manuscript (Page 10, Lines 239–240) and the Fig. 2 legends (Page 31).

Line 265: suggest to rephrase – this could be attributed to ...

[Response] We appreciate the reviewer’s comments. We have rephrased the sentence into “This could be in part a function of the continuous mutation and recombination abilities of norovirus, generating novel strains with high potential of causing outbreak events and sporadic cases” (Page 14, Lines 332–334).

Lines 289-290: 20% of all child deaths over which time period?

[Response] We appreciate the reviewer’s corrections and suggestions. We have added the time period in 2016 (Page 15, Line 358).

Line 297: if data available on water supply and sanitation services coverage over the study period, it would be an interesting aspect to consider in the analyses, in addition to the rural/urban categories.

[Response] We appreciate the reviewer’s valuable comments. This is a fantastic idea to relate the water supply and sanitation services coverage to the diarrhea incidence and enteropathogens detection. Unfortunately, these data are inaccessible at this moment, which point should be explored in the future investigation.

Lines 297-300: better understanding AMR patterns (not only among bacteria) is an important avenue for future research.

[Response] We agree with the reviewer that AMR is an important avenue for future research. We have added this comment to the revised manuscript “Better understanding Anti-Microbial Resistance patterns of enteropathogens is an important avenue for future research.” (Page 16, Lines 367–369).

Lines 309-313: it is unclear to me what the authors are suggesting here, concretely.

[Response] We have revised the sentence as “The current knowledge on whether the circulation of one pathogen enhances or diminishes the infection incidence of another might lead to an enhanced estimation of the diagnosis or treatment choice. The elaboration of such interactions may have economic implications through public health planning, and the clinical management of diarrhea disease.” (Page 16, Lines 378–383).

Lines 314-319: do the authors feel that their dataset is sufficient to propose a diagnostic tree to distinguish between bacterial, viral and parasitic infections among Chinese patients with acute diarrhoea? Otherwise, would using findings from their study to refine decision criteria for the selection of laboratory analyses to be performed on clinical samples be a more realistic recommendation?

[Response] We appreciate the reviewer’s important comments. Because the number of

samples that were tested for parasitic infections were too small to perform differential diagnosis between bacterial, viral and parasitic infection, we only used data from samples tested for both bacterial and viral pathogens (25,239 cases) to distinguish between bacterial and viral infections. The results had been added to the revised the manuscript (Pages 11–12, Lines 260–276):

“To attain a differential diagnosis between bacterial and viral infections with acute diarrhea, a binary eXtreme Gradient Boosting model was applied on 5,816 patients with viral single infections and 2,942 patients with bacterial single infections. In total, 11 valid variables regarding demographical and clinical symptoms/syndromes were entered into model, with season, age, mucous stool, vomiting, respiratory symptoms, mushy stool, bloody stool, watery stool, sex, fever, and neurologic symptoms observed. Among them, season was the most important predictor with a relative contribution of 37.46%, followed by age (16.73%). The contributions of mucous stool, vomiting and respiratory symptoms were slightly larger than 5%, and the contribution for each of the others was less than 5%. The area under curve of the model was 0.79 (95% confidence interval: 0.77–0.81), accuracy=74.14%. (Supplementary Fig. 4A and 4B). The mean accuracy of cross validation was 74.52% and the variance was 4.17%. Based on this multivariate logistic regression model, children patients of acute diarrhea with vomiting and respiratory symptoms occurred in cold season were shown to be likely caused by virus infections, while adult with mucous stool or bloody stool occurred in warm season were probably infected with bacteria (Supplementary Table 11).”

We also commented this finding in the discussion “These findings may help refine clinician decision criteria for the selection of laboratory analyses to be performed on clinical samples. The degree to which the cost and benefits of such an approach might reduce hospital cost would be a promising area of future research.” (Page 16, Lines 387–390).

Line 349: detected or tested?

[Response] We appreciate the reviewer’s comments. It should be “tested”, which has been modified in the revised manuscript (Page 18, Line 430).

A summary of the decision criteria used to decide which pathogens to test for in a given sample and whether they might have led to bias in the detection of certain pathogens would be helpful. Reporting the frequency of testing for each pathogen could also address this point.

[Response] We appreciate the reviewer’s valuable comments. There is no decision criteria used to decide which pathogens to test for in a given sample. Instead, the participating hospitals or laboratories had predesigned priority of testing pathogens according to their different test capacity and would test all samples for the predesigned pathogen list. On the other hand, the patients who received different panel of testing pathogens were highly comparable for their demography, thus revealing minor bias caused by this test strategy. Still, this is a limitation of our study, which had been discussed in the revised manuscript (Page 18, Lines 429–434). Also

as suggested by the reviewer, we have described these data in the main text (Page 8, Lines 163–168 for viral pathogen; Page 9, Lines 197–205 for bacterial pathogen; Page 10, Lines 228–237 for parasitic pathogens), and provided the frequency of testing for each pathogen in Supplementary Table 5 (Pages 13–14 in the Supplementary Information).

Supplementary Materials and Methods:

- Ref. 4 – the URL doesn't seem to be accessible. How was the rural/urban status defined for a given patient address?

[Response] We appreciate the reviewer's corrections and suggestions. We have checked the URL in Ref. 4. In this study, patients were classified into rural and urban depending on the administrative classification of their residence address. This information has been added to the revised Supplemental Material and Methods (Page 3 in the Supplementary Information).

Sup. Fig 3:

- related to my previous comment, were samples tested for parasites also tested for viruses and bacteria in some cases? If not, why?

[Response] We appreciate the reviewer's comments. Yes, some of the samples that were tested for parasites were also tested for viruses and bacteria. This has led to 3,330 patients who were tested for each of the 23 gastroenteritis pathogens. This is a limitation of our study, which has been discussed in the revised manuscript (Page 18, Lines 429–434).

- The figure appears in contradiction with the statement on page 2 of Supplementary Methods and Materials: "Parasites are not included in the study of pathogen co-infection patterns..."

[Response] We appreciate the reviewer's important comments. Co-infection patterns was only explored within parasites, not for parasites-bacteria or parasites-viruses. To clarify this confusion, we have revised this sentence as "Parasites are not studied for their interaction with virus or bacteria due to only small number of samples were tested for parasites." (Page 3 in the Supplementary Information).

- Bacterium (singular) and bacteria (plural) should be inverted on this figure.

[Response] Done as suggested.

Sup. Table 1:

- headings are unclear – what does "detect" mean here? Should it be replaced with "tested", as in Fig 3?

[Response] We appreciate the reviewer's corrections and suggestions. We have revised "detected" into "tested".

Suggest to include information on rural/urban categories in Supplementary Tables 1, 2, 3 if available.

[Response] We appreciate the reviewer's comments. We have provided information on rural/urban categories in Supplementary Table 1 (Page 7 in the Supplementary Information), Supplementary Table 3 (Page 10 in the Supplementary Information) and provided the comparison of positive rate in urban and rural patients with diarrhea, 2009–2018 in Supplementary Table 9 (Page 19 in the Supplementary Information).

Sup. Table 4: it seems like the line headings like “All viruses detected” should be replaced with “All viruses tested” (same for viruses, parasites).

[Response] We appreciate the reviewer's corrections and suggestions. We have revised “detected” into “tested”.

REVIEWERS' COMMENTS

Reviewer #1 (Remarks to the Author):

The authors have revised the paper entitled: "Etiological, epidemiological, and clinical features of acute diarrhea in China 2009-2018: an active sentinel surveillance study" to my satisfaction. Almost all the queries were addressed adequately.

There is just one section on page 15, Line 343-345 that still requires revision. This sentence should still be rephrased for clarity:

These findings provided clear information on the optimal timing of prevention strategies, e.g., for norovirus diagnosis and prevention, once a vaccine is available, should be a priority in the 2-3 year age group, and additionally at the age of five years.

Typographical errors:

Line 117 – patents.....should be.....patients

Line 193 – Please check this sentence: Among patients under 5 years of age, only the rate for rotavirus was significant sex ($p < 0.001$)

I suggest one of the following changes to clarify.

Among patients under 5 years of age, only the rate for rotavirus was significant ($p < 0.001$)

Among patients under 5 years of age, only the rate for rotavirus was significant for sex ($p < 0.001$)

A final check of the English in the paper would be beneficial.

Reviewer #2 (Remarks to the Author):

To the authors,

Thank you for this new version of the manuscript and your comprehensive response to review comments. I have no further comments other than a few notes regarding sentences that may need language clarification (but please note I am not a native English speaker).

Lines 111-113 : ..., 58,620 patients *were* tested *for* all seven viruses, 59,384 patients *were* tested *for* all 13 bacteria, 11,167 patients *were* tested *for* all three parasites, and 3,330 patients *were* tested *for* all 23 pathogens.

Lines 152-154: I would suggest to rephrase: In the three adult groups (18-45, ...), the top three pathogens were DEC, NTS and V. parahaemolyticus.

Lines 192-193: I believe the statement should be inverted (?) – "Among patients under 5 years old, only sex was significant for the rate of rotavirus detection (p...)."

Lines 281-283: I would suggest to use the simple past and split into two sentences for clarity - ... the daily temperature had significantly affected most of the detected viruses and bacteria, but with different directions of effects. Incidence ratios ...

Line 322: replace "began to decrease" with "decreasing"

Line 324: ... are the threshold*s* ...

Line 335: ... viral infection, ** divergent turnpoints of age were presented ...

Responses to the reviewers:

Reviewer #1 (Remarks to the Author):

The authors have revised the paper entitled: “Etiological, epidemiological, and clinical features of acute diarrhea in China 2009-2018: an active sentinel surveillance study” to my satisfaction. Almost all the queries were addressed adequately.

There is just one section on page 15, Line 343-345 that still requires revision. This sentence should still be rephrased for clarity:

[Response] We appreciate the reviewer’s corrections and suggestions.

We have rewritten this sentence as: “These findings of the effect of age on the pathogen detection might provide scientific evidence for finding the optimal timing to enhance prevention measures for each pathogen, e.g., for the diagnosis and prevention of norovirus, once a vaccine is available, it should be a priority in the 2–3 year age group, and additionally at the age of 15–16 years.” (Page 13, Lines 327–331)

Typographical errors:

Line 117 – patents.....should be.....patients

[Response] Thanks for the reviewer’s correction, and we have revised it as suggested (Page 5, Line 99).

Line 193 – Please check this sentence: Among patients under 5 years of age, only the rate for rotavirus was significant sex ($p < 0.001$)

I suggest one of the following changes to clarify.

Among patients under 5 years of age, only the rate for rotavirus was significant ($p < 0.001$)

Among patients under 5 years of age, only the rate for rotavirus was significant for sex ($p < 0.001$)

[Response] Thanks for the reviewer’s correction, and we have revised this sentence as suggested (Page 8, Lines 173–174).

A final check of the English in the paper would be beneficial.

[Response] We appreciate the reviewer’s comments. We also have a native English-speaker who has checked for clarity of expression throughout our manuscript.

Reviewer #2 (Remarks to the Author):

To the authors,

Thank you for this new version of the manuscript and your comprehensive response to review comments. I have no further comments other than a few notes regarding sentences that may need language clarification (but please note I am not a native English speaker).

[Response] We appreciate the reviewer's corrections and suggestions.

We have a native English-speaker who has checked for clarity of expression throughout our manuscript.

Lines 111-113 : ..., 58,620 patients *were* tested *for* all seven viruses, 59,384 patients *were* tested *for* all 13 bacteria, 11,167 patients *were* tested *for* all three parasites, and 3,330 patients *were* tested *for* all 23 pathogens.

[Response] Revised as suggested (Page 5, Lines 92–94).

Lines 152-154: I would suggest to rephrase: In the three adult groups (18-45, ...), the top three pathogens were DEC, NTS and V. parahaemolyticus.

[Response] Revised as suggested (Page 6, Lines 133–135).

Lines 192-193: I believe the statement should be inverted (?) – “Among patients under 5 years old, only sex was significant for the rate of rotavirus detection (p...)”

[Response] We appreciate the reviewer's comments. We have rephrased the sentence into “Among patients under 5 years of age, only the rate for rotavirus was significant for sex (p<0.001)” (Page 8, Lines 173–174).

Lines 281-283: I would suggest to use the simple past and split into two sentences for clarity - ... the daily temperature had significantly affected most of the detected viruses and bacteria, but with different directions of effects. Incidence ratios ...

[Response] Revised as suggested (Page 11, Line 266).

Line 322: replace “began to decrease” with “decreasing”

[Response] Revised as suggested (Page 12, Line 307).

Line 324: ... are the threshold*s* ...

[Response] Revised as suggested (Page 13, Line 309).

Line 335: ... viral infection, ** divergent turnpoints of age were presented ...

[Response] Revised as suggested (Page 13, Line 320).